# Miniature radiocarbon measurements (< 150 μg C) from sediments of Lake Żabińskie, Poland: effect of precision and dating density on age-depth models

Paul D. Zander[1], Sönke Szidat[2], Darrell S. Kaufman[3], Maurycy Żarczyński[4], Anna I. Poraj-Górska[4], Petra Boltshauser-Kaltenrieder[5], Martin Grosjean[1]

[1]Institute of Geography & Oeschger Centre for Climate Change Research, University of Bern, Bern, CH-3012, Switzerland
[2]Department of Chemistry and Biochemistry & Oeschger Centre for Climate Change Research, University of Bern, Bern, CH-3012, Switzerland
[3]School of Earth and Sustainability, Northern Arizona University, Flagstaff, AZ 86011, USA
[4]Faculty of Oceanography and Geography, University of Gdańsk, Gdańsk, 80-309, Poland
[5]Institute of Plant Sciences & Oeschger Centre for Climate Change Research, University of Bern, Bern, CH-3013, Switzerland

Correspondence to: Paul D. Zander (paul.zander@giub.unibe.ch)

**Abstract**

The recent development of the MIni CArbon DAting System (MICADAS) allows researchers to obtain radiocarbon ($^{14}$C) ages from a variety of samples with miniature amounts of carbon (< 150 μg C) by using a gas ion source input that bypasses the graphitization step used for conventional $^{14}$C dating with accelerator mass spectrometry (AMS). The ability to measure smaller samples, at reduced cost compared with graphitized samples, allows for greater dating density of sediments with low macrofossil concentrations. In this study, we use a section of varved sediments from Lake Żabińskie, NE Poland, as a case
study to assess the usefulness of miniature samples from terrestrial plant macrofossils for dating lake sediments. Radiocarbon samples analyzed using gas-source techniques were measured from the same depths as larger graphitized samples to compare the reliability and precision of the two techniques directly. We find that the analytical precision of gas-source measurements decreases as sample mass decreases, but is comparable with graphitized samples of a similar size (approximately 150 μg C). For samples larger than 40 μg C and younger than 6000 yr BP, the uncalibrated 1σ age uncertainty is consistently less than
150 years (± 0.010 F$^{14}$C). The reliability of $^{14}$C ages from both techniques is assessed via comparison with a best-age estimate for the sediment sequence, which is the result of an OxCal V-sequence that integrates varve counts with $^{14}$C ages. No bias is

evident in the ages produced by either gas-source input or graphitization. None of the $^{14}$C ages in our dataset are clear outliers; the 95% confidence intervals of all 48 calibrated $^{14}$C ages overlap with the median best-age estimate. The effects of sample mass (which defines the expected analytical age uncertainty) and dating density on age-depth models are evaluated via simulated sets of $^{14}$C ages that are used as inputs for OxCal P-sequence age-depth models. Nine different sampling scenarios were simulated in which the mass of $^{14}$C samples and the number of samples were manipulated. The simulated age-depth models suggest that the lower analytical precision associated with miniature samples can be compensated for by increased dating density. The data presented in this paper can improve sampling strategies and can inform expectations of age uncertainty from miniature radiocarbon samples as well as age-depth model outcomes for lacustrine sediments.

Keywords: radiocarbon, MICADAS, lake sediments, OxCal, age-depth modeling

## 1 Introduction

Radiocarbon ($^{14}$C) dating is the most widely used technique to date sedimentary sequences that are less than 50,000 years old. The robustness of age-depth models can be limited by the availability of suitable material for dating; this is particularly a problem for studies on sediments from alpine, polar, or arid regions where terrestrial biomass is scarce. Most accelerator mass spectrometry (AMS) labs recommend that samples contain 1 mg or more of carbon for reliable $^{14}$C age estimations. It is well established that terrestrial plant macrofossils are the preferred material type for dating lake sediments because bulk sediments or aquatic macrofossils may have an aquatic source of carbon, which can bias $^{14}$C ages (Groot et al., 2014; MacDonald et al., 1991; Tornqvist et al., 1992; Barnekow et al., 1998; Grimm et al., 2009). Furthermore, a high density of $^{14}$C ages (i.e. one age per 500 years) is recommended to reduce the overall chronologic uncertainty of age-depth models (Blaauw et al., 2018). Researchers working on sediments with low abundances of terrestrial plant macrofossils face difficult choices about whether to date sub-optimal materials (e.g. bulk sediment or aquatic macrofossils), pool material from wide sample intervals, or rely on few ages for their chronologies. The problem of insufficient material can affect age estimates at all scales from an entire sedimentary sequence to a specific event layer which a researcher wishes to determine the age of as precisely as possible.

Recent advances have reduced the required sample mass for AMS $^{14}$C analysis, opening new opportunities for researchers (Delqué-Količ et al., 2013; Freeman et al., 2016; Santos et al., 2007; Shah Walter et al., 2015). The recently developed MIni CArbon DAting System (MICADAS) has the capability to analyze samples with miniature masses via the input of samples in

a gaseous form, thus omitting sample graphitization (Ruff et al., 2007, 2010a, 2010b; Synal et al., 2007; Szidat et al., 2014;

Wacker et al., 2010a, 2010b, 2013). Samples containing as little as a few μg C can be dated using the gas-source input of the MICADAS. The analysis of such small samples provides several potential benefits for dating lake sediments: 1) the possibility to date sediments that were previously not dateable using [14]C due to insufficient material, 2) the ability to date sedimentary profiles with a greater sampling density and lower costs per sample, and 3) the ability to be more selective when selecting material to be analyzed for [14]C. The disadvantage of miniature samples is increased analytical uncertainty, which is a

consequence of lower counts of carbon isotopes and the greater impact of contamination on the measurement results. The goal of this study is to assess the potential benefits and limits of applying miniature [14]C measurements to dating lake sediments. We aim to answer the following questions in this study: 1) How reliable and how precise are gas-source [14]C ages compared with graphitized ages? 2) What is the variability of [14]C ages obtained from a single stratigraphic level? 3) How do analytical precision and dating density affect the accuracy and precision of age-depth models for lake sediments?

In this study, we use the sediments of Lake Żabińskie, Poland, as a case study to investigate the application of gas-source [14]C measurements to lake sediments. We focus on a continuously varved segment of the core, which spans from roughly 2.1 to 6.8 ka. We report the results of 48 radiocarbon measurements (17 using graphitization and 31 using the gas-source input) in order to compare the precision and reliability of gas-source [14]C ages with graphitized samples. The core was sampled such that up

to five ages were obtained from 14 distinct stratigraphic depths. A floating varve chronology was integrated with the [14]C ages to produce a best-age estimate using the OxCal V-sequence routine (Bronk Ramsey, 2008). This best-age estimate is used as a benchmark for the [14]C results. The results of our [14]C measurements were used to constrain a statistical model designed to simulate sets of [14]C ages in order to test nine different hypothetical sampling scenarios in which we manipulate the number of ages and the mass of C per sample, which determines the analytical uncertainty of the simulated ages. By comparing the results

of the simulated age-depth model outputs from these simulated [14]C ages with the best-age estimate from which the simulated ages were derived, we can improve our understanding of how the number of ages and their analytical precision influence the accuracy and precision of radiocarbon-based age-depth models.

## 2 Materials and Methods

### 2.1 Core material and radiocarbon samples

Cores were obtained from Lake Żabińskie (coring site: 54.1318° N, 21.9836° E, 44 m water depth) in 2012 using an UWITEC piston corer (90 mm diameter). Lake Żabińskie is a small (41.6 ha), relatively deep (44.4 m) kettle-hole lake located at an altitude of 120 m a.s.l. The catchment is 24.8 km$^2$ and includes two other smaller lakes: Lake Purwin and Lake Łękuk. Average temperatures range from 17 °C in summer to -2 °C in winter. Annual precipitation is 610 mm, with the annual peak in summer (JJA). The geology of the catchment is primarily glacial till, sandy moraines and glacial fluvial sands and gravels (Szumański, 2000). Modern land cover in the catchment is a mixture of cultivated fields and primarily oak-lime-hornbeam and pine forests (Wacnik et al., 2016). The high relative depth (6.1%; calculated according to Wetzel et al., 1991) of Lake Żabińskie leads to strong seasonal stratification, bottom-water anoxia, and the preservation of varves in the sediments (Bonk et al., 2015a, 2015b; Tylmann et al., 2016; Żarczyński et al., 2018). Varve-based chronologies and $^{14}$C measurements have been published for the most recent 2000 years of the Lake Żabińskie sedimentary sequence (Bonk et al., 2015a; Żarczyński et al., 2018). These studies show major changes to varve structure and a three-fold increase in sedimentation rates in response to increased cultivation and deforestation, beginning around 1610 CE. Prior to this time, land cover in the region was relatively stable, with forest/woodland cover dominating the landscape from the early Holocene until the 17$^{th}$ century CE (Wacnik, 2009; Żarczyński et al., 2019).

A composite sediment profile was constructed from overlapping, 2-m-long cores by correlating distinctive stratigraphic features. The composite sequence spans 19.4 m. Published downcore varve counts stop above a ~90-cm-thick slump/deformed unit. This slump event is dated to 1962-2071 cal yr BP (present = 1950 CE) based on an extension of the varve count published in Żarczyński et al., 2018. This study focuses on a section of core (7.3-13.1 m depth in our composite sequence) directly below this slump unit; this section was selected because it features continuous well-preserved varves throughout the section. Samples of 1- to 2-cm-thick slices of sediment were taken from the core (sample locations and core images are found in Supplementary File 1), then sieved with a 100 μm sieve. Macrofossil remains were identified and photographed (Supplementary File 2), and only identifiable terrestrial plant material was selected for $^{14}$C measurements. Suitable macrofossils from a single stratigraphic level were divided into subsamples for analysis, with the goal of producing one graphitized $^{14}$C age and 2-4 gas-source ages from each depth. When convenient, we grouped samples by the type of material (leaves, periderm, needles, seeds or woody scales), though 11 samples are a mixture of material types. In most cases, subsamples within a stratigraphic level are assumed to be independent, meaning they may have different true ages. However, there are some subsamples that were taken from

single macrofossil fragments (six subsamples taken from two fragments sampled from two different depths), thus these samples have the same true age. It is also possible that subsamples from a single depth may be from the same original material without our knowledge (i.e. a macrofossil could break into several pieces while sieving, and these pieces could be analyzed as separate subsamples).

Sample material was treated with an acid-base-acid (ABA) method at 40°C, using 0.5 mol/L HCl, 0.1 mol/L NaOH and 0.5 mol/L HCl for 3 h, 2 h and 3 h, respectively. After drying at room temperature, samples were weighed, and those less than 300 µg were input to the gas ion source via combustion in an Elementar Vario EL Cube elemental analyser (Salazar et al., 2015). Larger samples were graphitized following combustion using automated graphitization equipment (AGE) (Szidat et al., 2014).

Radiocarbon data was processed using the software BATS (Wacker et al., 2010a). Additional corrections were applied to the data to account for cross contamination (carryover), and constant contamination (blanks) (Gottschalk et al., 2018; Salazar et al., 2015). The parameters for these corrections were calculated based on standard materials (the primary NIST standard oxalic acid II (SRM 4990C) and sodium acetate (Sigma-Aldrich, No. 71180) as $^{14}$C-free material) run with the sample batches. We applied a constant contamination correction of $1.5 \pm 0.2$ µg C with $0.72 \pm 0.11$ F$^{14}$C and a cross contamination correction of

$(1.2 \pm 0.3$ %$)$ from the previously run sample. Measurement uncertainties were fully propagated for each correction. In total, 48 ages were obtained from 14 distinct stratigraphic levels (17 graphitized and 31 gas-source measurements).

## 2.2 Varve count

Varves in Lake Żabińskie are biogenic, with calcite-rich pale laminae deposited in spring and summer, and darker laminae containing organic detritus and fine clastic material deposited in winter (Żarczyński et al., 2018). We defined the boundary of

125 each varve year by the onset of calcite precipitation (i.e., the upper boundary of dark laminae and lower boundary of light-colored laminae). Varves were counted using CooRecorder software (Larsson, 2003) on core images obtained from a Specim PFD-CL-65-V10E linescan camera (Butz et al., 2015). Three people performed independent varve counts, and these three counts were synthesized, and uncertainties calculated according to the methodology recommended by Żarczyński et al. (2018) yielding a master varve count with asymmetric uncertainties.

Because of the slump deposit above our section of interest, the varve chronology is 'floating' and must be constrained by the $^{14}$C ages. Several different approaches were used to compare the varve count with the $^{14}$C ages, all of which rely on some assumptions. One method is to tie the varve count to the radiocarbon based age at a chosen depth in the core. We tested this

method using the median calibrated age of the uppermost dated level as the tie point. Such an approach assumes that the radiocarbon-based age at the tie point is correct. An additional drawback is that the choice of tie-point is arbitrary and can change the resulting varve count ages. Alternatively, we used least squares minimization to fit the varve count to all radiocarbon ages (Hajdas et al., 1995) by minimizing the offset between the varve count and the combined calibrated radiocarbon age at each dated level. However, we focus on a third, more sophisticated method, which is the OxCal 4.3 V-sequence (Bronk Ramsey, 2008, 2009; Bronk Ramsey and Lee, 2013). This technique integrates all available chronological information including varve counting and $^{14}C$ ages into a single model to determine a best-age estimate for the sequence (see sect. below for more details). The advantages of this approach are that all ages are considered equally likely to be correct (or incorrect), and the error estimate of the V-sequence is relatively consistent along the profile, whereas the error associated with the varve count is small at the top of the section, but increases downcore. Additionally, this technique allows for the possibility that the master varve count is incorrect (within the expected uncertainty of the count).

## 2.3 Age-depth modeling

Age-depth modeling was performed using OxCal 4.3 (Bronk Ramsey, 2008, 2009; Bronk Ramsey and Lee, 2013), which integrates the IntCal13 calibration curve (Reimer et al., 2013) for $^{14}C$ ages with statistical models that can be used to construct age-depth sequences. As an initial test to compare the reliability of gas-source ages and graphitized ages, and their effect on age-depth models, we produced three P-sequence models: one using all obtained $^{14}C$ ages, one using only graphitized ages, and one using only gas-source ages. For all OxCal models in this study, ages measured from the same depth were combined (using the function *R_combine*) into a single $^{14}C$ age with uncertainty before calibration and integration into the age-depth sequence. This choice was verified by the chi-squared statistic calculated by OxCal to test the agreement of ages sampled from a single depth. For every combination of ages except one, we find that the chi-squared test is passed at 0.05 significance level. We justify the use of the combine function even for the grouping that failed to pass the chi-squared test (samples from 811 cm depth) because all ages in this group overlap, and there is no significant difference when models are run with the ages separated at this depth (less than 5 years difference for median age, and CI). The OxCal P-sequence uses a Bayesian approach in which sediment deposition is modelled as a Poisson (random) process. A parameter (k) determines the extent to which sedimentation rates are allowed to vary. For all P-sequence models in this study, we used a uniformly distributed prior for k such that $k_0 = 1$, and $\log_{10}(k/k_0) \sim U(-2, 2)$; this allows k to vary between 0.01 and 100. Sediment deposition sequences are constrained by likelihood functions produced by the calibration of radiocarbon ages. Thousands of iterations of sediment deposition sequences

are produced using Markov Chain Monte Carlo (MCMC) sampling (Bronk Ramsey, 2008). These iterations can then be summarized into median age estimates, with confidence intervals.

The varve counts and all [14]C ages were incorporated into an OxCal V-sequence in an approach similar to that used by Rey et al. (2019). The V-sequence differs from the P-sequence in that it does not model sediment deposition. Instead, the V-sequence uses 'Gaps' (the amount of time between two points in a sequence) to constrain the uncertainty of radiocarbon ages. The Gap can be determined from independent chronological information such as varve counts or tree ring counts. We input the number of varves in 10 cm intervals to the V-sequence as an age 'Gap' with associated uncertainty. The OxCal V-sequence assumes normally distributed uncertainties for each gap, whereas our varve count method produces asymmetric uncertainty estimates. We used the mean of the positive and negative uncertainties as inputs to the V-sequence. However, OxCal sets the minimum uncertainty of each 'Gap' equal to 5 years, which in most cases is larger than the mean uncertainty in our varve count over a 10 cm interval. By including the varve counts as an additional constraint, the V-sequence produces a more precise age-depth relation than the P-sequence, which only considers the radiocarbon ages.

## 2.4 Age-depth model simulation

In order to test the effects of analytical uncertainty and dating density (number of ages per time interval) on age-depth models, we designed an experiment in which nine different sampling scenarios were simulated for the Lake Żabińskie sedimentary sequence to determine the expected precision and accuracy of resulting age-depth models. Three different sampling densities were simulated for the 5.8-m-long section: 5 ages, 10 ages, and 20 ages (equivalent to approximately 1, 2, and 4 ages per millennium, respectively). For each of these sampling densities three different sample-size scenarios were simulated: 35 μg C, 90 μg C, 500 μg C. These scenarios were designed to represent different sampling circumstances such as high or low abundances of suitable material for [14]C analysis, and different budgets for [14]C analysis. Radiocarbon ages were simulated using a technique similar to Trachsel and Telford (2017). In brief, we distributed the simulated samples evenly by depth across the 5.8-m-long section, and then used the median output of the OxCal V-sequence as the assumed true age for a given depth. This calibrated assumed true age was back-converted to [14]C years using IntCal13 (Reimer et al., 2013). A random error term was added to the [14]C age to simulate the analytical uncertainty. The error term was drawn from a normal distribution with mean zero and standard deviation equivalent to the age uncertainty determined from the relationship between sample mass and age uncertainty found in the results of our [14]C measurements (Fig. 1A). The same expected analytical uncertainty was used for the age uncertainty for each simulated age. For a sample with 35 μg C, we expect a measurement uncertainty of ± 148 years (or ±

0.0114 F$^{14}$C), which is representative for the average age of all samples in this study (approximately 4000 $^{14}$C yr BP). In reality, older samples would have greater age uncertainty, while younger samples would have less uncertainty. However, the effect of these differences on the performance of simulated age-depth models would be minimal as roughly half the ages would be more precise and half would be less precise. These simulated $^{14}$C ages were input into an OxCal P-sequence using the same uniform distribution for the k-parameter as described in the previous section. This experiment was repeated 30 times for each scenario to assess the variability of possible age-model outcomes. We quantify the accuracy of the age-depth models as the deviation of the median modelled age from the best-age estimate at a given depth. We define precision as the width of the age-depth model confidence interval (CI).

## 3 Results

### 3.1 Radiocarbon measurements

In total, 48 radiocarbon measurements on terrestrial plant macrofossils were obtained from the section of interest yielding values from 0.475 – 0.777 F$^{14}$C (2030 to 5990 $^{14}$C yr BP; Table 1). Thirty-one ages were measured using the gas-source input; these samples contained between 11 and 168 µg C. Seventeen samples containing between 115 and 691 µg C were measured using graphitization. Analytical uncertainties for the $^{14}$C measurements range from ± 0.0027 to ± 0.0306 F$^{14}$C (± 41 to ± 328 years) with higher values associated with the smallest sample masses. The uncertainties for gas-source measurements and graphitized measurements are comparable for samples that contain a similar amount of carbon (Fig. 1). Samples containing less than 40 µg C (roughly equivalent to 80 µg of dry plant material) produce uncertainties greater than ± 150 years (1σ). We use a power-model fit with least-squares regression, to estimate the typical age uncertainty for a given sample mass (r$^2$ = 0.90, p < 0.001, Fig. 1). The resulting power model is nearly identical to what would be expected based on the assumed Poisson distribution of the counting statistics where the uncertainty follows the relationship N$^{-0.5}$ (N = the number of measured $^{14}$C atoms).

When comparing measurements taken from within a single sediment slice we find good agreement for all $^{14}$C ages, regardless of whether the samples were analyzed with the gas-source input or via a graphitized target (Fig. 2), and no clear bias based on the type of macrofossil that was dated (Fig. 3). One method to test whether the scatter of ages is consistent with the expectations of the analytical uncertainty is a reduced chi-squared statistical test, also known as Mean Square Weighted Deviation (MSWD) in geochronological studies (Reiners et al., 2017). If the spread of ages is exactly what would be expected from the analytical

uncertainty, the value of this statistic is 1. Lower values represent less scatter than expected, and larger values represent more scatter than expected. Of the 11 sampled depths with three or more ages, only one (811 cm, MSWD = 3.07) returned an MSWD that exceeds a 95% significance threshold for acceptable MSWD values that are consistent with the assumption that the age scatter is purely the result of analytical uncertainty.

## 3.2 Varve count and age-depth modeling

In total, 4644 (+155/- 176) varves were counted in the section of interest, with a mean varve thickness of $1.26 \pm 0.58$ mm (Fig. 4). Full varve count results are available at https://dx.doi.org/10.7892/boris.134606. Sedimentation rates averaged over 10 cm intervals range from 0.91 to 2.78 mm/year. All chronological data ($^{14}$C ages and varve counts) were integrated to generate a best-age estimate for the section of interest using an OxCal V-sequence (output of the Oxcal V-sequence is available at https://dx.doi.org/10.7892/boris.134606). This produced a well-constrained age-depth model with a 95% confidence interval (CI) width that ranges from 69 to 114 years (mean 86 years). OxCal uses an agreement index to assess how well the posterior distributions produced by the model (modelled ages at the depth of $^{14}$C ages) agree with the prior distributions (calibrated $^{14}$C ages). The overall agreement index for our OxCal V-sequence is 66.8%, which is greater than the acceptable index of 60%. Three of the fourteen dated levels in the V-sequence had agreement indices less than the acceptable value of 60% (A = 22.8, 48.5, 52.6% for sample depths = 1283.0, 1176.1, 732.5 cm, respectively), nonetheless we find the model fit acceptable as all 48 $^{14}$C ages overlap with the median output of the V-sequence. We use the V-sequence as a best-age estimate for subsequent data comparisons and analyses. Alternative methods of linking the floating varve count with $^{14}$C ages confirm that the $^{14}$C ages are consistent with the varve count results. When the varve count is tied to the combined radiocarbon ages at the uppermost dated level (732.5 cm), we find that all other radiocarbon ages overlap with the varve count when considering the uncertainty of the varve count. If least squares minimization is used to minimize the offset between all radiocarbon ages and the varve count, we again find that all radiocarbon ages overlap with the master varve count (without considering varve count uncertainty). The result from the least squares minimization technique is highly similar to the OxCal V-sequence output.

To test the reliability of gas-source ages versus graphitized ages we created three OxCal P-sequences using: 1) all $^{14}$C ages, 2) only graphitized ages, and 3) only gas-source ages. The results of all three of these age-depth models agree well with the best-age estimate of the V-sequence, although with larger 95% CIs (Fig. 2). The agreement index was greater than the acceptable value of 60 for all three models overall, and for each dated depth within all three models. The P-sequence using all $^{14}$C ages spans $4838 \pm 235$ years, which is slightly greater than, but overlapping with, the total number of varves counted (the V-

sequence estimates 4681 ± 79 years in the section). There is no clear bias observed in the age-depth models produced using either the gas-source or graphitized samples. The P-sequence outputs clearly show that a very precise age can narrowly constrain the age-model uncertainty at the depth of that sample, however, if dating density is low, the uncertainty related to interpolation between ages becomes large. Despite the lower precision of the gas-source ages, the model based on only gas-source ages actually has a lower mean CI width than the model with graphitized ages (mean 95% CI width: 373 years for the gas-source model, 438 years for the graphitized model). However, a direct comparison between the gas-source-only and the graphitized-only age models is confounded by differences in the number and spacing of samples. Specifically, there are no graphitized ages between the top of the section (724 cm) and 811 cm, and between 1082 and 1200 cm, which results in wide CI in these sections. On the other hand, uncertainty is reduced compared to the gas-source model in the depths adjacent to the graphitized ages due to higher precision such that 40% of the section (in terms of depth) has lower age uncertainty in the graphitized model.

## 3.3 Age-depth model simulations

Nine different sampling scenarios (described in Sect. 2.3) were simulated to test the effects of dating density and analytical precision on age-depth model confidence intervals. For each of the nine scenarios, sets of $^{14}$C ages were simulated 30 times to create an ensemble of age-depth models for each scenario. One set of these simulated age-depth models is shown in Fig. 5, and an animation of the full set of simulated models is available online (Supplementary File 3). The age-depth models were evaluated for their precision (mean width of the 95% CI) and accuracy (the mean absolute deviation from the best-age estimate; summarized in Fig. 6 and Table 2). As expected, we find that increased dating density and increased sample masses improve both the accuracy and precision of the age-depth models. It is notable that increasing the number of ages can compensate for the greater uncertainty associated with smaller sample sizes. For instance, the mean CI of age-depth models based on ten, 90 µg C samples is narrower than age-depth models with five, 500 µg C samples (Table 2). However, the effect of analytical precision is greater on the mean absolute deviation from the best-age estimate. Increased dating density does tend to reduce the deviation from the best-age estimate (especially if the ages are imprecise), but the three scenarios that use 500 µg samples perform better than all other scenarios, in terms of deviation from the best-age estimate, regardless of the sampling density. Additionally, increased dating density does not improve the deviation from the best-age estimate for the 500 µg sample scenarios. This result may be due to the relatively constant sedimentation rates in our sedimentary sequence, which reduces errors caused by interpolation in scenarios with low dating density. Another prominent pattern in the simulations is the large

spread of performance for models with relatively few and imprecise ages (Fig. 6). Increasing the number of samples and, especially, the mass of samples has a large impact on the agreement among the different iterations of each scenario.

An additional measure of age-model quality is the Chron Score rating system (Sundqvist et al., 2014), which does not assess age-depth model fit, rather it assesses the quality of inputs used to generate an age-depth model. Thus the Chron Score provides an assessment of the 9 sampling scenarios that is independent of the choice of age-depth modelling software, or parameter selection during age-depth model construction. The Chron Score is calculated from three criteria used to assess the reliability of core chronologies: 1) delineation of downcore trend (D), 2) quality of dated materials (Q), and 3) precision of calibrated ages (P). These metrics are combined using a reproducible formula to provide a Chron Score (G) in which higher values represent more reliable chronologies:

$$G = -w_D D + w_Q Q + w_P P$$

We used the default weighting parameters ($w_D$, $w_Q$, and $w_P = 0.001$, 1 and 200) for each component of the Chron Score formula as described in Sundqvist et al. (2014). The quality (Q) parameter depends on two factors – the proportion of ages which are not rejected or reversed (i.e. an older age stratigraphically above a younger age), and a qualitative classification scheme for material types. We modified the threshold for determining if an age is considered a reversal such that if a $^{14}$C age is older than a stratigraphically higher age by more than the age uncertainty ($1\sigma$), the age is considered to be stratigraphically reversed. This is different from the default setting, which is 100 years. For the material type classification (m), the simulated age models were assigned the value 4, which is the value assigned to chronologies based on terrestrial macrofossils. For more details on the Chron Score calculation see Sundqvist et al. (2014). The mean Chron Scores for the simulated age models (Table 2) show that doubling dating density substantially improves the Chron Score, but the effect is greater when moving from 5 to 10 ages than from 10 to 20 ages. The effect of increased precision on the Chron Score is also substantial; it is essentially defined by the Chron Score formula, in which precision is assessed as $P = s^{-1}$ where s is the mean 95% range of all calibrated $^{14}$C ages. The effect of precision on the Chron Score is also determined by the weighting factors mentioned above.

## 4 Discussion

### 4.1 Radiocarbon measurements

The results of our [14]C measurements from repeated sampling of single stratigraphic levels provide useful information for other researchers working with miniature [14]C analyses, or any [14]C samples from lake sediments. We show that there is an exponential relationship between sample mass and the resulting analytical uncertainty (Fig. 1). We use the relationship shown in Fig. 1A to define the age uncertainty of our simulated ages, however it is important to note that this relationship is only valid for samples with a similar age to the samples in this study (approx. 2000-7000 cal yr BP). Older samples will yield greater age uncertainty for the same mass of C due to fewer [14]C isotopes (Gottschalk et al., 2018). The measurement uncertainty in F[14]C units is not affected by age (Fig. 1B). The exact parameters of these relationships will also depend on laboratory conditions, however, the general shape of the relationship is valid. These data can inform researchers about the expected range of uncertainty for [14]C ages from samples of a given size. We find that samples larger than 40 μg C yield ages that are precise enough to be useful for dating Holocene lake sediments in most applications, and even smaller samples can provide useful ages if no other material is available.

It is well documented that [14]C ages can be susceptible to sources of error that are not included within the analytical uncertainty of the measurements. Such errors can be due to lab contamination, sample material which is subject to reservoir effects (i.e. bulk sediments or aquatic organic matter; Groot et al., 2014; MacDonald et al., 1991; Tornqvist et al., 1992), or from depositional lags (terrestrial organic material which is older than the sediments surrounding it; Bonk et al., 2015; Howarth et al., 2013; Krawiec et al., 2013). Errors related to reservoir effects can be avoided by selecting only terrestrial plant material for dating (Oswald et al., 2005). Floating or shoreline vegetation should also be avoided as these plants may uptake $CO_2$ released by lake degassing (Hatté and Jull, 2015). Dating fragile material such as leaves (as opposed to wood) may reduce the chances of dating reworked material with a depositional lag, but generally this source of error is challenging to predict and depends on the characteristics of each lake's depositional system. To identify ages affected by depositional lags, it is necessary to compare with other age information. Consequently, the identification of outlying ages is facilitated by increased dating density.

In our dataset, multiple [14]C measurements were performed on material taken from a single layer, which enables outlier detection. We find that the scatter of [14]C ages obtained from the same depths is generally consistent with what would be

expected based on the analytical uncertainties of the ages. There are no clear outliers in the data; every single [14]C age has a calibrated 95% CI that overlaps with the median of our best-age estimate OxCal V-sequence (and this result is confirmed by alternative methods of linking the varve count to [14]C ages). This agreement between the varve count and the [14]C ages is evidence that no age in this dataset is incongruent with the other available chronological information (other [14]C ages and varve counts). This notion is further demonstrated by the fact that 10 of 11 sampled levels from which we obtained three or more ages returned an MSWD within the 95% confidence threshold for testing age scatter (see Sect. 3.1; Reiners et al., 2017). This test is typically used for repeated measurements on the same sample material, however, in our study, many of the measurements from within a single sediment slice are from material that has different true ages. The MSWD test indicates that the variability in ages among samples from within a single sediment slice can reasonably be expected given the analytical uncertainty. However, in this study, no more than five samples were measured per depth, and thus the range of acceptable values for the MSWD is relatively wide due to the small number of degrees of freedom. Additionally, the analytical uncertainties are relatively large for the gas-source samples, allowing for wide scatter in the data without exceeding the MSWD critical value. Despite these caveats, the consistency between the variability among ages from one level and the analytical uncertainties allows us to make two important conclusions. 1) The analytical precision estimates are reasonable, even for miniature gas-source samples. 2) When material is carefully selected and taxonomically identified for dating, the sources of error that are not considered in the analytical uncertainty (e.g. contamination or depositional lags) are relatively minor in our case study. However, this second conclusion is highly dependent on the sediment transport and depositional processes, which are site specific. Depositional lags still likely have some impact on our chronology. Six [14]C ages from plant material collected from the Lake Żabińskie catchment in 2015 yielded a range of ages from 1978-2014 CE (Bonk et al., 2015) suggesting that the assumption that [14]C ages represent the age of the sediments surrounding macrofossils is often invalid. The scale of these age offsets is likely on the scale of a few decades for Lake Żabińskie sediments, which is inconsequential for many radiocarbon-based chronologies, but is the same order of magnitude as the uncertainty of our best-age estimate from the OxCal V-sequence, and should be considered when reporting or interpreting radiocarbon-based age determinations with very high precision.

The lack of outliers in our dataset is an apparent contrast with the findings of Bonk et al. (2015), who report that 17 of 32 radiocarbon samples taken from the uppermost 1000 years of the Lake Żabińskie core were outliers. The outlying ages were older than expected based on the varve chronology, and this offset was attributed to reworking of terrestrial plant material. The identification of outliers did not take into account uncertainties of the radiocarbon calibration curve and varve counts, which could explain some of the differences between the [14]C and the varve ages. Still, 8 of 32 ages reported by Bonk et al. (2015)

have calibrated 2σ age ranges that do not overlap with varve count age (including the varve count uncertainty). The higher outlier frequency in the Bonk et al. (2015) data might be explained by their generally more precise ages and the fact that their varve count is truly independent from the [14]C ages.

Additionally, our dataset allows us to compare the results of [14]C ages obtained from different types of macrofossil materials, which we grouped into the following categories: leaves (including associated twigs), needles, seeds, periderm, woody scales, and samples containing mixed material types (Fig. 3). When comparing the calibrated median age of each sample to the median of our best-age estimate, we find that the difference between the age offsets of the different material types is not significant at the $\alpha = 0.05$ level (ANOVA, F = 2.127, p = 0.08). This is likely due to our selective screening of sample material, which only includes terrestrial plant material while avoiding aquatic insect remains or possible aquatic plant material, as well as the relatively small number of samples within each material type. There does appear to be a tendency for seeds to produce younger ages, and two of the three woody scale samples yielded ages that are approximately 300 years older than the best-age estimate. This could be due to the superior durability of woody materials compared with other macrofossil materials, which enables wood to be stored on the landscape prior to being deposited in the lake sediments. A larger number of samples would allow for more robust conclusions about the likelihood of certain material types to produce biased ages.

## 4.2 The OxCal V-sequence best-age estimate

In this study we have tested multiple approaches to assigning absolute ages from [14]C ages to a floating varve count (Fig. 4). Using a single tie-point relies on a potentially arbitrary selection of tie-point location and yields large uncertainty intervals when considering both the varve count uncertainty and the uncertainty of calibrated ages. Using least squares minimization of the offset between all radiocarbon ages and the varve count has the advantage of using all the [14]C ages rather than one tie-point, however this approach does not consider varve count uncertainties and does not directly yield an estimate of uncertainty derived from the radiocarbon age uncertainties. The OxCal V-sequence is unique in that all age information is integrated into a statistical framework including the probability functions of [14]C ages and the uncertainty associated with the varve count as well. In contrast to the other two approaches, the V-sequence can change the total number of years in the sequence compared to the original varve count. However, the addition of 37 years in the V-sequence is well within the uncertainty of the varve count (+155/- 176). The V-sequence approach is expected to provide more precise and more reliable age estimates than either varve counting or radiocarbon-based age models alone. The resulting age-depth relation has a relatively narrow CI (mean 95% CI is 86 yr). Extremely precise age estimates were also produced using this method for Moossee, Switzerland by Rey et al.

(2019). A combination of varve counts and $^{14}$C ages from the Moossee sediments generated a V-sequence output with a mean 95% CI of 38 years. The higher precision in the Moossee study compared to our V-sequence output is primarily attributed to the higher dating density in Moosse with 27 radiocarbon ages over ~3000 years (3.9-7.1 ka) versus our study, which used 48 ages, but from only 14 unique depths, over ~4700 years. This comparison shows that repeated measurements from the same

depth are less useful than analyses from additional depths. This approach to integrating varve counts and $^{14}$C ages could potentially be improved by a better integration of varve count uncertainties into the OxCal program. Currently the uncertainties on age 'Gaps' in OxCal are assumed to be normally distributed and cannot be less than 5 years. Nevertheless, the result of the OxCal V-sequence is an age-depth model that is much more precise than those constructed only using $^{14}$C ages and provides a useful reference to compare with the $^{14}$C ages. It is important to note that the best-age estimate is not independent of the $^{14}$C

ages; it is directly informed by the $^{14}$C ages.

### 4.3 Age-depth model simulations

The simulated age-depth modelling experiment allows us to assess the effects of dating density and sample mass (expected precision) on the outputs of age-depth models constructed for the section of interest in the Lake Żabińskie sediment core. Models based on relatively few, but very precise ages, are tightly constrained at the sample depths, but the CI widens further

away from these depths (Fig. 5, Supplementary File 3). In contrast, models based on a greater sampling density produce confidence intervals with relatively constant width. If models are built using a high density of imprecise ages, the CI of the model output can actually be narrower than the CI of the individual ages. Bayesian age-depth models in particular can take advantage of the stratigraphic order of samples to constrain age-depth models to be more precise than the individual ages that make up the model (Blaauw et al., 2018), however this is only achievable when dating density is high enough. The results

from this experiment suggest that, in the case of the Lake Żabińskie sequence, doubling the number of ages can approximately compensate for an increased analytical uncertainty of 50 years.

The choice of OxCal to produce age-depth models from these hypothetical sampling scenarios may have some influence on the results, however we expect that the key findings are replicable for any Bayesian age-depth model routine (i.e. Bacon or

405 Bchron; Blaauw and Christen, 2011; Haslett and Parnell, 2008). To demonstrate this, we used Bacon (Blaauw and Christen, 2011, 2018) to generate age-depth models for one iteration of the simulated sampling scenarios, and compared the results to those generated by OxCal. We find that the Bacon-generated models are highly similar to the OxCal models, and the patterns

observed in terms of model precision and accuracy are reasonable similar to those obtained from Oxcal models. The Bacon results can be found in Supplementary File 4.

The Chron Score results provide a succinct summary of the reliability of the chronologies produced in the different simulated sampling scenarios and is independent of model selection. The Chron Score becomes more sensitive to changes in precision as precision increases, so the difference in the Chron Scores between the 500 μg and 90 μg scenarios (1σ uncertainty of ± 39 and 92 years, respectively) is greater than the difference between the 90 μg and 35 μg scenarios (1σ uncertainty of ± 92 and 415 148 years, respectively). Increased dating density consistently improves the Chron Score results, with a stronger impact seen when shifting from 5 to 10 ages compared to shifting from 10 to 20 ages. The improvement of the Chron Score due to increased dating density is generally consistent for each of the different sample mass scenarios. This differs from the age-depth model statistics where increased dating density has a greater impact on precision in the larger sample mass scenarios (more precise ages). The opposite effect is seen in the mean absolute deviation results, where mean absolute deviation is reduced substantially 420 as dating density increases for the smaller sample scenarios, and not at all for the 500 μg scenario. For all measures of chronologic performance, we find a greater improvement when increasing the number of ages from 5 to 10 ages compared to increasing from 10 to 20 ages, suggesting there are some diminishing returns from increased dating density. This result is in accordance with the results of Blaauw et al. (2018). While the Chron Score results are dependent on the parameters chosen for the calculation, they intuitively make sense. Because Chron Score results use only the simulated $^{14}$C ages as input and are 425 unaffected by the age modelling routine, the patterns exhibited in the scores may be more applicable to a variety of sedimentary records.

In real-world applications, there are additional advantages from increasing dating density. Many lacustrine sequences have greater variability in sedimentation rates than the sequence modelled here. More fluctuations in sedimentation rate require a 430 greater number of ages to delineate the changes in sedimentation. Additionally, outlying ages and age scatter beyond analytical uncertainty are not considered in this modelling experiment. In most cases, detecting outlying ages becomes easier as dating density increases. Because this experiment is only applied to a single sedimentary sequence, the results may not be directly applicable for other sedimentary records with different depositional conditions. In the future, this type of age model simulation could be applied to a range of sedimentary sequences with a variety of depositional conditions.

## 4.4 Recommendations for radiocarbon sampling strategy

Radiocarbon sampling strategies will always be highly dependent on project-specific considerations such as how the chronology will affect the scientific goals of the project, budget and labor constraints, the nature of the sedimentary record in question, and the availability of suitable materials. A goal of this study is to provide data that can inform sampling strategies for building robust chronologies, particularly in cases where suitable material may be limited. Firstly, an iterative approach to $^{14}$C measurements is preferred. An initial batch of measurements should target a low dating density of perhaps one date per 2000 years. Subsequent samples should aim to fill in gaps where age uncertainty remains highest (Blaauw et al., 2018), or where preliminary age-depth trends appear to be non-linear. In accordance with many previous studies (e.g. Howarth et al., 2013; Oswald et al., 2005), we advocate for careful selection of material identified as terrestrial in origin. If the mass of such material is limited, the MICADAS gas-source is useful for dating miniature samples, and we are convinced that miniature samples of terrestrial material are preferable to dating questionable material or bulk sediments. Samples as small as a few µg C can be measured using the MICADAS, though samples larger than 40 µg C are recommended for more precise results (mid to late Holocene samples containing 40 µg C are expected to have analytical uncertainty of ~138 years). Dating small amounts of material from single depths is also preferable to pooling material from depth segments that may represent long time intervals. A general rule of thumb is to avoid taking samples with depth intervals representing more time than the expected uncertainty of a $^{14}$C age. To improve the accuracy of age-depth models, a higher priority should be placed on achieving sufficiently high dating density (ideally greater than one age per 500 years; Blaauw et al., 2018) using narrow sample-depth intervals. In most cases, this goal should be prioritized over the goal of gathering larger sample masses in order to reduce analytical uncertainties. The results of this study and others (e.g. Blaauw et al., 2018; Trachsel and Telford, 2017) clearly indicate that increased sampling density improves the accuracy, precision and reliability of age-depth models.

Multiple measurements from within a single stratigraphic depth, as we have done in this study, can be useful in sediments where age scatter (possibly from reworked material) is expected. In such cases, multiple measurements from a single depth could allow for identification of certain types of material that should be avoided. If age scatter is not expected, single measures of pooled macrofossils are more cost-effective than repeat measurements from a single depth.

Although increased dating density does incur greater cost, gas-source ages have lower costs compared to graphitized ages allowing for greater dating density at similar cost. Injecting $CO_2$ into the AMS rather than generating graphite and packing a

target substantially reduces the effort to analyze a sample following pre-treatment, and additionally reduces some chance of contamination during graphitization. These advantages are partly offset by additional operator attention required during gas source measurements. How these differences translate to per-sample costs depends on the pricing structures implemented in each lab. Cost estimates from two MICADAS labs at the University of Bern and Northern Arizona University range between approximately 15 and 33% lower costs for gas-source measurements compared to graphitized samples. Use of smaller samples can also reduce the labor time required to isolate suitable material from the sediment, however handling and cleaning miniature samples can add additional challenges which increases labor time.

**5 Conclusions**

- AMS [14]C analysis of Holocene terrestrial plant macrofossils using the MICADAS gas-ion source produces unbiased ages with similar precision compared to graphitized samples that contain similar mass of carbon (approximately 120-160 µg C).

- The precision of a [14]C age can be approximately estimated based on the amount of carbon within a sample. Holocene samples containing greater than 40 µg C produce [14]C measurements with analytical uncertainty expected to be less than $\pm 0.01$ F[14]C (150 years for samples than are approximately 4000 years old). Uncertainty increases exponentially as samples get smaller so 10 µg C samples are expected to have uncertainty of $\pm 0.021$ F[14]C (277 years).

- The variability among ages obtained from 1- or 2-cm-thick samples in the Lake Żabińskie sediment core is compatible with the variability expected due to analytical uncertainty alone.

- We find no clear evidence in our dataset for age bias based on the type of macrofossil material dated, which we limited to terrestrial plant material.

- Judging from the output of age-depth models, the lower precision of miniature gas-source ages can be compensated for by increasing sampling density. Based on sets of simulated [14]C ages that mimic the [14]C ages of our study core, together with age-depth models generated using OxCal, doubling dating density roughly compensates for a decrease in analytical precision of 50 years.

- The effect of [14]C age precision is among several factors that influence chronological precision. The thickness of the depth interval used to obtain samples, the ability to select identifiable terrestrial materials or to analyze more than one type of material, the reliability of detecting age outliers, and the amount of variability in sedimentation rate all determine the accuracy and precision of an age-depth model, which are both improved by increasing the number of ages.

•    This study can inform sampling strategies and provide expectations about radiocarbon-based age-depth model outcomes.

## Data Availability

The key datasets associated with this manuscript (Varve count results and the best-age estimate OxCal V-sequence output) are available at the Bern Open Repository and Information System, https://dx.doi.org/10.7892/boris.134606

## Author Contributions

PZ prepared samples, designed and performed the age modelling experiment, analyzed results, and prepared the manuscript with contributions from all authors. MG, DK, SS, and PZ designed the strategy and goals of the study. MZ, APG, and PZ performed varve counting. PBK identified and selected suitable macrofossils. SS oversaw $^{14}$C analyses. DK assisted with laboratory work.

## Competing interests

The authors declare no competing interests

## Acknowledgements

Core materials were supplied by Wojciech Tylmann, University of Gdańsk. Edith Vogel and Gary Salazar assisted with $^{14}$C sample measurements. This project was funded by Swiss National Science Foundation grants 200021_172586 and IZSEZO-180887 and by the Polish National Science Centre grant 2014/13/B/ST10/01311.

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

 **Figure and Table Captions**

**Figure 1: A) Age uncertainty of AMS radiocarbon ages (without calibration) versus the mass of carbon in the sample. Note that these samples date to approximately 2000-6000 BP; older ages will have greater age uncertainties. Note the logarithmic scale on the x-axis. The black line represents the best-fit power model for our dataset. B) Same as A, except uncertainties are plotted as measurement uncertainty in $F^{14}C$ units. This measure of uncertainty is not directly influenced by the age of the sample.**

**Figure 2: A) Comparison of age-depth model outputs from OxCal (Bronk Ramsey, 2008, 2009; Bronk Ramsey and Lee, 2013; Reimer et al., 2013). From left to right: OxCal V-sequence using all $^{14}C$ ages as well as varve counts as inputs; OxCal P-sequence using all $^{14}C$ ages as inputs; OxCal P-sequence using only gas-source $^{14}C$ ages; OxCal P-sequence using only graphitized $^{14}C$ ages. The median age of the V-sequence is considered the best-age estimate and is repeated in all four panels as a red line. Gray lines represent the upper and lower limits of the 95% confidence interval of each model. Black lines represent the median ages of the P-sequences. B) Radiocarbon calibrated age probability density functions for each measured age, grouped by composite depth. The best-age estimates from the OxCal V-sequence are plotted as red lines for comparison. The = symbol adjacent to some probability density functions indicates that these ages (within a single depth) came from the same specimen and have the same true age.**

**Figure 3: Offsets between median calibrated $^{14}C$ ages and the best age estimate from the OxCal V-sequence. Data are grouped by material type. Higher values indicate that the sample age is older than the best-age estimate.**

**Figure 4: All radiocarbon ages and their 95% calibrated uncertainties plotted with the varve count results. The gray bands show the varve count tied to the combined calibrated age of the uppermost $^{14}C$ ages (at 732.5 cm) with dark grey representing the uncertainty calculated from the three replicated varve counts and light gray representing the uncertainty of the tie point. Dashed green is the varve count fit to the $^{14}C$ ages using least squares minimization of the offset between the varve age and the combined $^{14}C$ ages at each sampled depth.**

**Figure 5: Results of age-model simulations to test the effects of sampling density and sample mass on age-model results. Each panel shows the output of an OxCal P-sequence using simulated $^{14}C$ ages as inputs compared with the best-age estimate from the V-**

sequence (shown in red). Simulated $^{14}$C ages are based on the decalibrated best-age estimate of a given depth and the expected uncertainty associated with the mass C in the simulated $^{14}$C age, which defines not only the age uncertainty, but also a random error term added to each simulated age. Plots show one ensemble member out of 30 simulations. An animation of all 30 simulations can be found in Supplementary File 3.

Figure 6: A) Boxplots showing the distribution of the mean 95% confidence interval widths produced by simulated age-depth models. Results are grouped by dating density along the x-axis, and by sample mass (smaller mass = greater uncertainty) using different colors. Each boxplot represents the distribution of results produced for 30 unique sets of simulated $^{14}$C samples. Data points that are greater (less) than the 75th (25th) percentile plus (minus) 1.5 times the interquartile range are plotted as single points beyond the extent of the whiskers. B) Same as A, but showing the mean absolute deviation from the best-age estimate (median output of OxCal V-sequence).

Table 1: Results of the 48 $^{14}$C analyses obtained for this study. Uncertainties of $^{14}$C ages refer to 68% probabilities (1σ) whereas ranges of calibrated and modelled ages represent 95% probabilities.

Table 2: Table summarizing the effect of dating density (number of ages) and analytical precision (sample mass) on the accuracy, precision and reliability of OxCal P-sequence models generated from simulated $^{14}$C ages. Each of the nine scenarios was simulated 30 times; presented values are the mean of the 30-member ensemble. Precision is assessed by the mean width of the age-depth model 95% confidence interval. Accuracy is measured by the mean absolute deviation from the OxCal V-sequence best-age estimate, which is the reference from which $^{14}$C ages were simulated. Chron Score is a metric designed to assessing the reliability of age-depth models where higher numbers represent greater reliability (Sundqvist et al., 2014).

**Supplementary Files**

Supplementary File 1: Core images and location of $^{14}$C ages.

Supplementary File 2: Microscope images of macrofossils used for $^{14}$C dating.

**Supplementary File 3: Animation of OxCal P-sequence age-depth models for all 30 iterations of simulated sampling scenarios (animated version of Figure 4).**

**Supplementary File 4: Comparison of Bacon and OxCal simulated age-depth models**

Figure 1

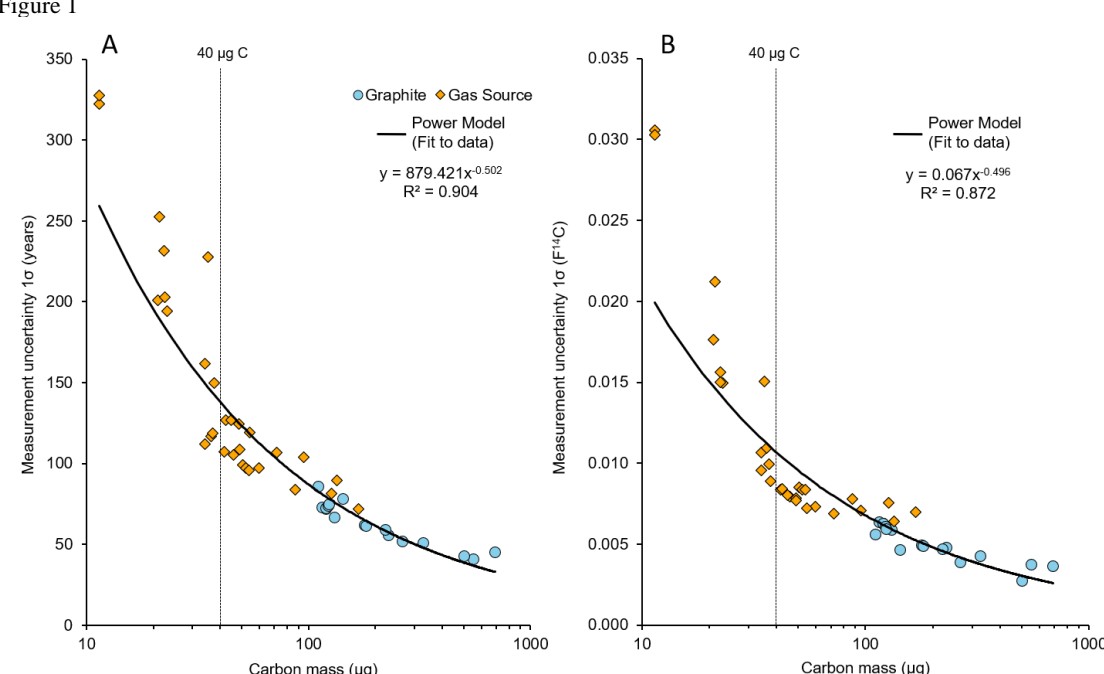

Figure 2

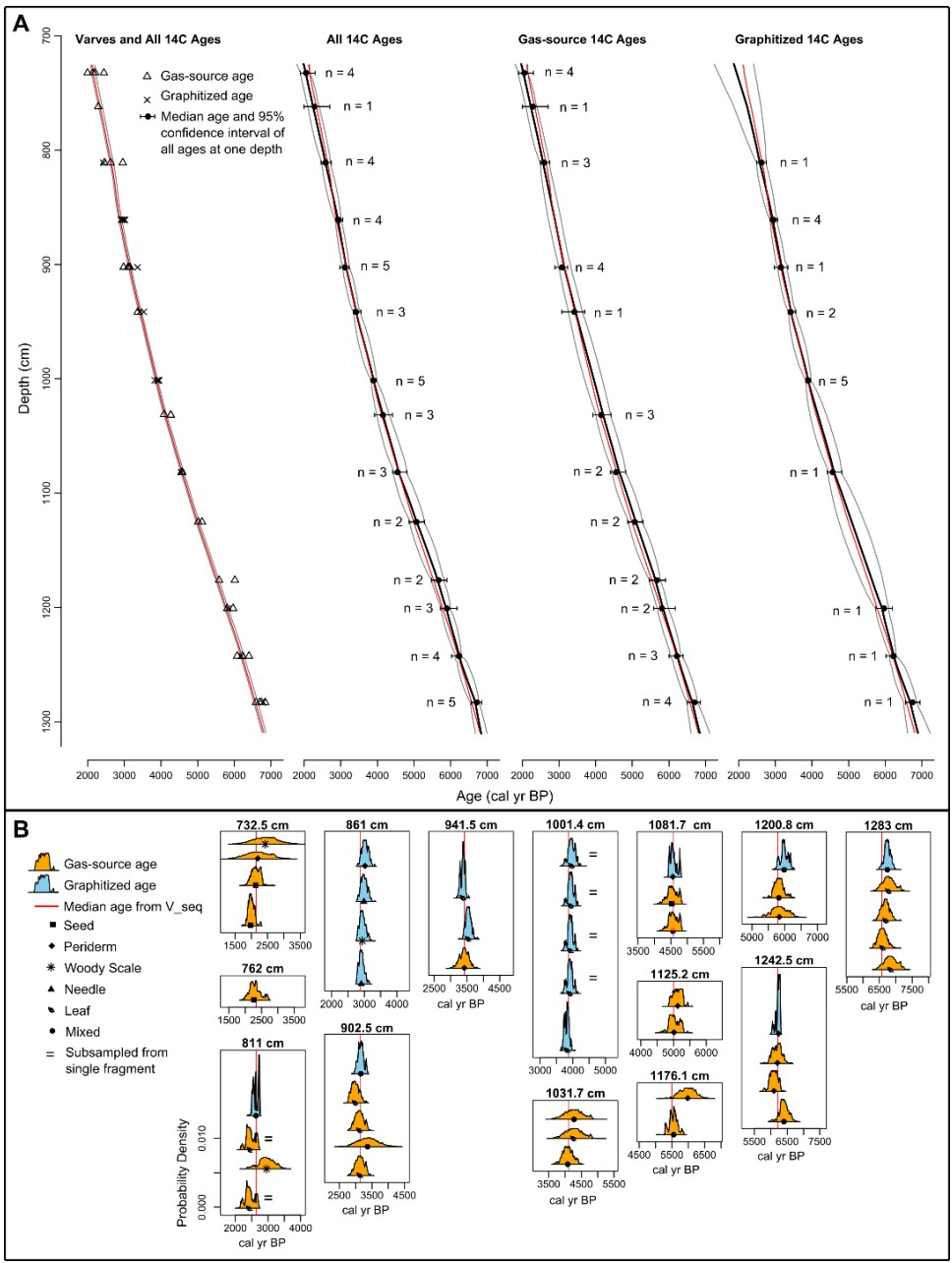

Figure 3

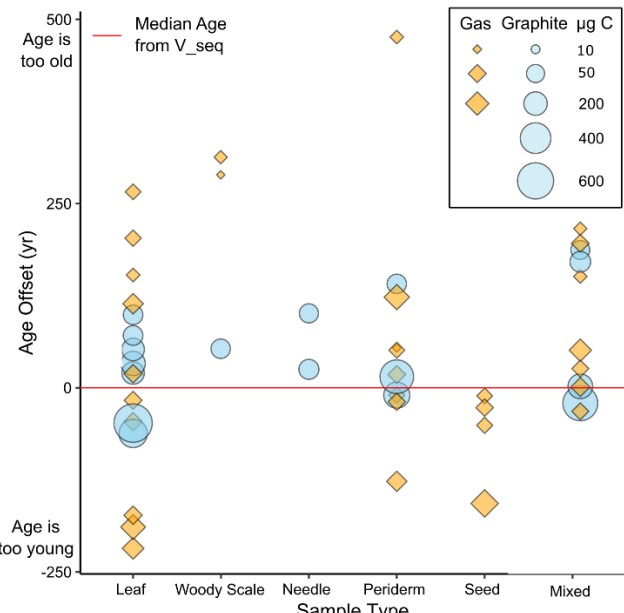

Figure 4

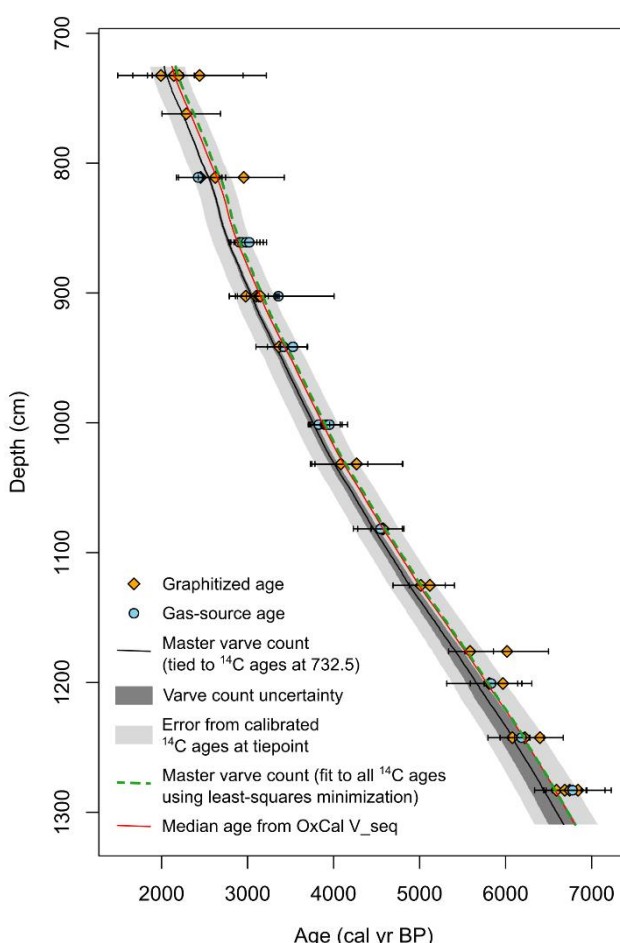

Graphitized age
Gas-source age
Master varve count
(tied to $^{14}$C ages at 732.5)
Varve count uncertainty
Error from calibrated
$^{14}$C ages at tiepoint
Master varve count (fit to all $^{14}$C ages
using least-squares minimization)
Median age from OxCal V_seq

Figure 5

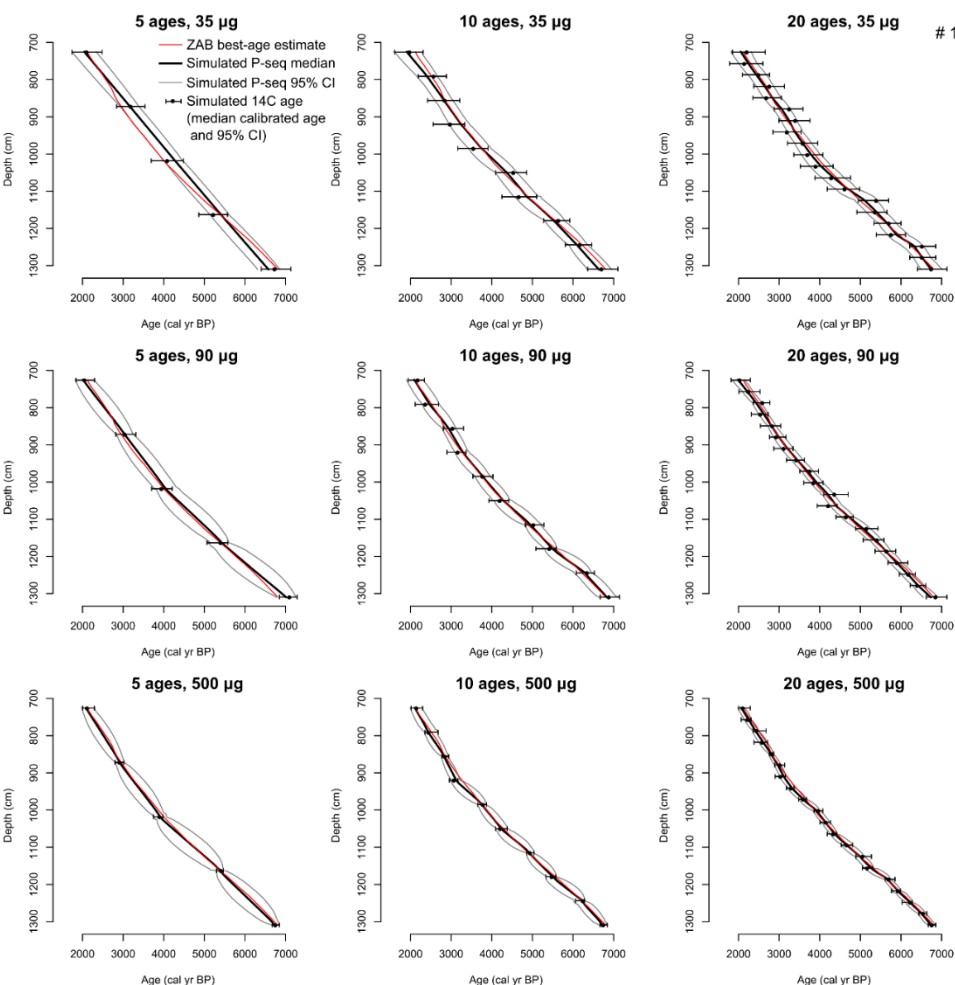

Figure 6

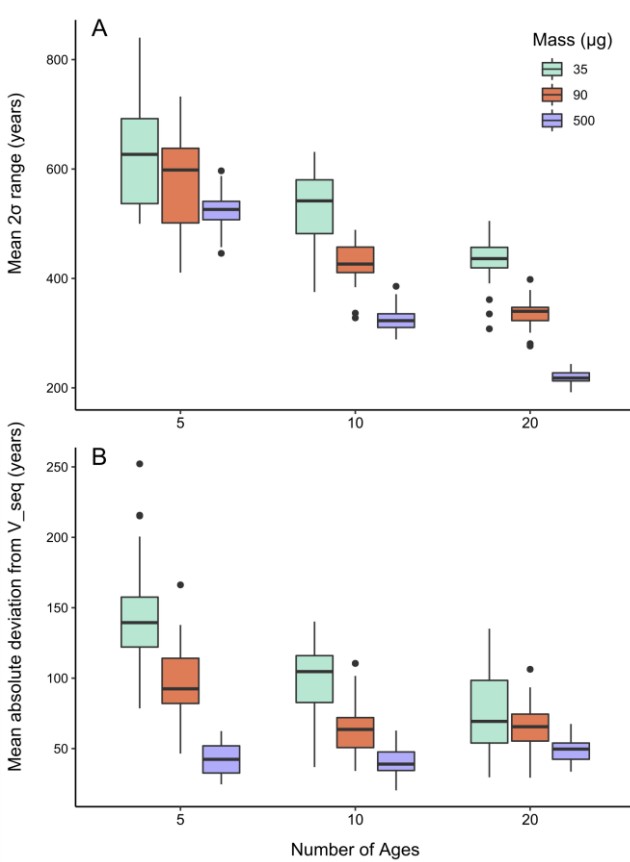

Table 1

| Lab ID | Core ID | Top Core Depth (cm) | Bottom Core Depth (cm) | Centered Composite Depth (cm) | Carbon Mass (μg) | Gas/ Graphite | $^{14}$C age (BP) | Calibrated Age (Cal yr BP)[1] | Modelled Age from OxCal V-sequence (Cal yr BP)[2] | Material |
|---|---|---|---|---|---|---|---|---|---|---|
| BE-9791.1.1 | ZAB-12-4-3-2 | 75 | 77 | 732.5 | 168 | Gas | 2028 ± 72 | 1823-2293 | 2106-2218 | *Pinus sylvestris* seed fragments (seed wing, and fragments of seed) |
| BE-9793.1.1 | ZAB-12-4-3-2 | 75 | 77 | 732.5 | 34 | Gas | 2149 ± 112 | 1867-2361 | 2106-2218 | Terrestrial seed fragment |
| BE-9792.1.1 | ZAB-12-4-3-2 | 75 | 77 | 732.5 | 11 | Gas | 2190 ± 322 | 1416-2968 | 2106-2218 | Periderm (coniferous) |
| BE-9794.1.1 | ZAB-12-4-3-2 | 75 | 77 | 732.5 | 11 | Gas | 2386 ± 328 | 1636-3325 | 2106-2218 | Woody scale |
| BE-9503.1.1 | ZAB-12-3-4-2 | 36 | 37 | 762 | 36 | Gas | 2273 ± 117 | 1998-2702 | 2297-2402 | *Alnus* seed fragments |
| BE-9502.1.2 | ZAB-12-3-4-2 | 85 | 86 | 811 | 87 | Gas | 2358 ± 84 | 2159-2715 | 2611-2703 | Dicotyledonous leaf fragment[3] |
| BE-9502.1.1 | ZAB-12-3-4-2 | 85 | 86 | 811 | 127 | Gas | 2379 ± 82 | 2183-2722 | 2611-2703 | Dicotyledonous leaf fragment[3] |
| BE-9501.1.1 | ZAB-12-3-4-2 | 85 | 86 | 811 | 21 | Gas | 2809 ± 201 | 2437-3447 | 2611-2703 | Deciduous tree/shrub woody scales |
| BE-9500.1.1 | ZAB-12-3-4-2 | 85 | 86 | 811 | 553 | Graphite | 2544 ± 41 | 2490-2754 | 2611-2703 | Dicotyledonous leaf fragments, woody scales |
| BE-9497.1.1 | ZAB-12-4-4-2 | 20 | 21 | 861 | 131 | Graphite | 2799 ± 67 | 2760-3076 | 2850-2929 | *Pinus sylvestris* needle |
| BE-9498.1.1 | ZAB-12-4-4-2 | 20 | 21 | 861 | 120 | Graphite | 2820 ± 72 | 2774-3143 | 2850-2929 | Woody scale |
| BE-9496.1.1 | ZAB-12-4-4-2 | 20 | 21 | 861 | 115 | Graphite | 2857 ± 73 | 2790-3174 | 2850-2929 | *Pinus sylvestris* needle |
| BE-9499.1.1 | ZAB-12-4-4-2 | 20 | 21 | 861 | 120 | Graphite | 2885 ± 72 | 2807-3229 | 2850-2929 | Periderm (deciduous) |
| BE-9495.1.1 | ZAB-12-4-4-2 | 61.5 | 62.5 | 902.5 | 21 | Gas | 3158 ± 252 | 2764-3984 | 3113-3187 | Periderm, Dicotyledonous leaf fragments, woody scales |
| BE-9494.1.1 | ZAB-12-4-4-2 | 61.5 | 62.5 | 902.5 | 54 | Gas | 2845 ± 96 | 2761-3215 | 3113-3187 | Dicotyledonous leaf fragment |
| BE-9494.1.2 | ZAB-12-4-4-2 | 61.5 | 62.5 | 902.5 | 50 | Gas | 2968 ± 99 | 2876-3374 | 3113-3187 | Dicotyledonous leaf fragment |
| BE-9494.1.3 | ZAB-12-4-4-2 | 61.5 | 62.5 | 902.5 | 52 | Gas | 2944 ± 97 | 2866-3358 | 3113-3187 | Dicotyledonous leaf fragment |

| | | | | | | | | | | |
|---|---|---|---|---|---|---|---|---|---|---|
| BE-9493.1.1 | ZAB-12-4-4-2 | 61.5 | 62.5 | 902.5 | 230 | Graphite | 2980 ± 56 | 2979-3340 | 3113-3187 | Dicotyledonous leaf fragments, periderm fragments |
| BE-9491.1.1 | ZAB-12-4-4-2 | 100.5 | 101.5 | 941.5 | 37 | Gas | 3197 ± 119 | 3078-3700 | 3391-3462 | Periderm |
| BE-9490.1.2 | ZAB-12-4-4-2 | 100.5 | 101.5 | 941.5 | 123 | Graphite | 3296 ± 74 | 3375-3696 | 3391-3462 | Dicotyledonous leaf fragments |
| BE-9490.1.1 | ZAB-12-4-4-2 | 100.5 | 101.5 | 941.5 | 328 | Graphite | 3145 ± 51 | 3226-3466 | 3391-3462 | Dicotyledonous leaf fragments |
| BE-9489.1.1 | ZAB-12-3-5-2 | 44 | 45 | 1001.4 | 691 | Graphite | 3542 ± 45 | 3697-3965 | 3845-3915 | Dicotyledonous leaf fragments |
| BE-9489.1.2 | ZAB-12-3-5-2 | 44 | 45 | 1001.4 | 179 | Graphite | 3593 ± 62 | 3717-4084 | 3845-3915 | Dicotyledonous leaf fragment[3] |
| BE-9489.1.4 | ZAB-12-3-5-2 | 44 | 45 | 1001.4 | 222 | Graphite | 3603 ± 59 | 3724-4086 | 3845-3915 | Dicotyledonous leaf fragment[3] |
| BE-9489.1.3 | ZAB-12-3-5-2 | 44 | 45 | 1001.4 | 182 | Graphite | 3616 ± 62 | 3725-4141 | 3845-3915 | Dicotyledonous leaf fragment[3] |
| BE-9489.1.5 | ZAB-12-3-5-2 | 44 | 45 | 1001.4 | 124 | Graphite | 3631 ± 75 | 3721-4153 | 3845-3915 | Dicotyledonous leaf fragment[3] |
| BE-9795.1.1 | ZAB-12-4-5-1 | 24 | 26 | 1031.2 | 42 | Gas | 3724 ± 107 | 3829-4417 | 4084-4155 | *Betula* seed fragments, terrestrial woody material, woody scale, periderm fragments |
| BE-9487.1.1 | ZAB-12-4-5-1 | 25 | 26 | 1031.7 | 23 | Gas | 3856 ± 194 | 3731-4832 | 4084-4155 | Leaf fragments |
| BE-9488.1.1 | ZAB-12-4-5-1 | 25 | 26 | 1031.7 | 22 | Gas | 3856 ± 203 | 3725-4836 | 4084-4155 | Wood fragment, Periderm fragments |
| BE-9485.1.1 | ZAB-12-4-5-1 | 75 | 76 | 1081.7 | 60 | Gas | 4062 ± 97 | 4296-4837 | 4540-4616 | Periderm, woody scales |
| BE-9486.1.1 | ZAB-12-4-5-1 | 75 | 76 | 1081.7 | 46 | Gas | 4042 ± 105 | 4249-4832 | 4540-4616 | *Betula alba* seed |
| BE-9484.1.1 | ZAB-12-4-5-1 | 75 | 76 | 1081.7 | 266 | Graphite | 4065 ± 52 | 4421-4813 | 4540-4616 | Periderm fragments |
| BE-9483.1.2 | ZAB-12-4-5-1 | 118.5 | 119.5 | 1125.2 | 49 | Gas | 4387 ± 108 | 4655-5318 | 4960-5042 | Periderm fragments |
| BE-9483.1.1 | ZAB-12-4-5-1 | 118.5 | 119.5 | 1125.2 | 135 | Gas | 4475 ± 90 | 4860-5434 | 4960-5042 | Periderm fragments |
| BE-9481.1.1 | ZAB-12-5-6-1 | 54 | 55.5 | 1176.1 | 95 | Gas | 4850 ± 104 | 5321-5887 | 5500-5591 | Woody seed fragments, leaf fragments, woody scales |
| BE-9482.1.1 | ZAB-12-5-6-1 | 54 | 55.5 | 1176.1 | 22 | Gas | 5246 ± 232 | 5485-6536 | 5500-5591 | Periderm fragments |
| BE-9480.1.1 | ZAB-12-5-6-1 | 79 | 80 | 1200.8 | 35 | Gas | 5081 ± 228 | 5320-6315 | 5745-5832 | Periderm fragments |
| BE-9479.1.1 | ZAB-12-5-6-1 | 79 | 80 | 1200.8 | 42 | Gas | 5063 ± 127 | 5586-6178 | 5745-5832 | Periderm, woody scale |

| BE-9478.1.1 | ZAB-12-5-6-1 | 79 | 80 | 1200.8 | 111 | Graphite | 5197 ± 86 | 5745-6190 | 5745-5832 | Periderm fragments and woody scales |
| BE-9476.1.1 | ZAB-12-5-6-2 | 5 | 6 | 1242.5 | 49 | Gas | 5601 ± 125 | 6032-6718 | 6175-6267 | Periderm and woody scale |
| BE-9475.1.1 | ZAB-12-5-6-2 | 5 | 6 | 1242.5 | 72 | Gas | 5294 ± 107 | 5768-6300 | 6175-6267 | Periderm |
| BE-9477.1.1 | ZAB-12-5-6-2 | 5 | 6 | 1242.5 | 45 | Gas | 5410 ± 127 | 5920-6439 | 6175-6267 | Periderm fragments |
| BE-9474.1.1 | ZAB-12-5-6-2 | 5 | 6 | 1242.5 | 504 | Graphite | 5402 ± 43 | 6020-6294 | 6175-6267 | *Pinus* periderm fragments |
| BE-9473.1.3 | ZAB-12-5-6-2 | 45.5 | 46.5 | 1283 | 34 | Gas | 5988 ± 162 | 6479-7250 | 6531-6643 | Dicotyledonous leaf fragments |
| BE-9473.1.2 | ZAB-12-5-6-2 | 45.5 | 46.5 | 1283 | 55 | Gas | 5787 ± 119 | 6317-6880 | 6531-6643 | Dicotyledonous leaf fragments |
| BE-9473.1.1 | ZAB-12-5-6-2 | 45.5 | 46.5 | 1283 | 74 | Gas | 5868 ± 107 | 6415-6949 | 6531-6643 | Dicotyledonous leaf fragments |
| BE-9473.1.4 | ZAB-12-5-6-2 | 45.5 | 46.5 | 1283 | 38 | Gas | 5936 ± 150 | 6436-7165 | 6531-6643 | Dicotyledonous leaf fragments |
| BE-9472.1.1 | ZAB-12-5-6-2 | 45.5 | 46.5 | 1283 | 143 | Graphite | 5916 ± 78 | 6547-6946 | 6531-6643 | Dicotyledonous leaf fragments, periderm fragment |

[1] Ages calibrated using OxCal 4.3 with the IntCal13 calibration curve (Bronk Ramsey, 2009; Reimer et al., 2013). The range reported here represents the 95% confidence interval.

[2] Range represents 95% confidence interval.

[3] These samples were subsampled from a single fragment prior to analysis, thus samples within the same depth with this symbol have the same true age.

705 Table 2

| Sample Mass (μg) | Expected Uncertainty (yr)[1] | Expected Uncertainty (F$^{14}$C) | Number of ages in model | | |
|---|---|---|---|---|---|
| | | | 5 ages (1.07 per kyr) | 10 ages (2.14 per kyr) | 20 ages (4.27 per kyr) |
| | | | *Mean 95% CI width (yr)* | | |
| 35 | ± 148 | ± 0.011 | 633 | 527 | 433 |
| 90 | ± 92 | ± 0.007 | 577 | 430 | 335 |
| 500 | ± 39 | ± 0.003 | 524 | 325 | 219 |
| | | | *Mean absolute deviation from OxCal V-sequence (yr)* | | |
| 35 | ± 148 | ± 0.011 | 144 | 99 | 78 |
| 90 | ± 92 | ± 0.007 | 98 | 64 | 65 |
| 500 | ± 39 | ± 0.003 | 42 | 40 | 49 |
| | | | *Chron Score* | | |
| 35 | ± 148 | ± 0.011 | 2.46 | 3.14 | 3.48 |
| 90 | ± 92 | ± 0.007 | 2.87 | 3.64 | 4.09 |
| 500 | ± 39 | ± 0.003 | 3.92 | 4.74 | 5.18 |

[1] Expected age uncertainty for an approximately 4000-year-old sample used to inform age-depth model simulations