# Peer review of "Miniature radiocarbon measurements (< 150 $\mu g$ C) from sediments of Lake Żabińskie, Poland: effect of precision and dating density on agedepth models"

_Geochronology, 2019_

## Referee Comment (RC1) · Anonymous Referee #1 · 4 Feb 2020

SUMMARY The manuscript describes the results of a case study in which radiocarbon ages obtained using gas-source technique are compared with radiocarbon ages of the conventional graphitized samples; both types of samples come from a number of selected depth intervals in a lake sediment core. Because this core supposedly has a relatively well resolved varve-based chronology (albeit floating and not shown in this manuscript), the authors integrate varve counts and two types of radiocarbon ages into a simulated 'best age estimate' model. They further demonstrate a series of exercises in generating the synthetic age-depth distributions with a purpose of illustrating the

effect of sampling density and sample size (mass carbon) on age model precision. According to the authors, the main idea of the work is an evaluation of how reliable gas source radiocarbon ages on miniature samples are for constructing age models. This is important for those lake records which lack enough datable material for the conventional radiocarbon analyses. The implications highlighted by the authors include (a) how to "improve sampling strategies" (the more age determinations the better, just as one may expect); and (b) what are the "expectations of age uncertainty". Among the benefits of skipping the graphitization step when using gas-source technique the authors cite "reduced cost", but there is no comparison provided for the respective costs for the two types of the techniques used.

NOTES The manuscript leans excessively toward theoretical evaluations of 'how things would be if...' and misses a discussion of several key points, which are named but not explored: depositional lags, outlier dates, examples of sample size effect on the radiocarbon date uncertainty as applied to a real core. This happens because the authors chose to (a) treat all their dates as equally good/likely; (b) use the 'best age estimate' for the sequence using everything at once, that is, they combined 3 varve count series + miniature+ regular + graphitized + gas source radiocarbon dates to make a single 'best age estimate'. No wonder there are no outliers if all these things are bundled together. As a reader, going from the Introduction to Discussion I expected to see the Figures showing step by step how overlapping varve-based chronologies look like first and how their cumulative error changes with depth, then how a certain number of graphitized regular ages help improving these chronologies and errors, and then how adding gas-source ages on the regular-size samples improves this chronology further, and then how adding gas-source ages on the less reliable miniature samples may or may not improve it even further. Instead, I see a single red line as 'best age estimate' from the very start and then 9 software-generated arbitrary age-depth scenarios. One does not need a sediment core to generate these latter graphs. Depositional lags for organic fragments are discussed in a purely theoretical way. There appear to be three different varve chronologies, why not show each one of them and see which

dates support which one (if any)? A test for potential age outliers would be more robust in this case. Supposedly, as admitted by the authors, the younger the portion of the studied sequence, the more robust is the varve-based chronology. Why not take advantage of this and have a closer look at the potential depositional lags in the most reliable upper portion of the record? What if the varve-only age models were used to compare with gas only and/or graphite only ages? The importance of mass for the reliability of the dates is stressed a number of times by the authors, but their Figures are not informative enough to illustrate this. For example, when discussing age offsets, why not show symbols of different size somehow proportionate to sample mass in Figure 3 and provide respective error bars for each of the dating points? If the sample mass is so important for the age date and bigger is definitely better (as shown in Fig.1), then is it really a good approach to consider all the dates equal in constructing the 'best age estimate'? If the authors found room for nine simulated graphs in the manuscript, I think it would be beneficial to see two-three age-depth graphs using best dates, small-sample dates, and then all dates for comparison. The section 4.4 "Recommendations..." is a disappointment as it states a number of trivial basic things about radiocarbon dating, which can be found anywhere and which are not supported by the data the authors present. For example: "we are convinced... that miniature samples... are better than bulk" – convinced based on what? There is no data presented to support this level of certainty. Indeed, it would have been a really nice test if they were to analyze at least couple bulk samples from the same horizons to see how they compare with those on sieved fragments. "Dating small amounts... is preferable to pooling ...", "a rule of thumb is...." – again, there are no data in the paper supporting this conclusion. It appears that these didactic statements are pasted from elsewhere. "If ages do not agree well ... youngest ae is most likely to be correct" - what about applying this principle to their own data set and showing how it works out in their studied portion of the lake record? It appears that in the paper the authors cite, Bonk et al. (2015) did just that and identified a number of outliers. Finally, the argument of 'cost reduction' for gas-source ages as compared to graphitized samples is used a

number of times in the manuscript. Indeed, costs are lab-specific, however, it would be of interest to have at least some estimate in % since the authors repeatedly bring this issue up themselves. I suggest substantial revisions, not "major" but at the same time not "minor" or technical either.

---

## Editor Comment (EC1) · Christine Hatté (Editor) · 19 Feb 2020

This paper intends to show that a chronology of a sequence is all the better constrained the more dates it contains. It also aims at showing that even on the basis of a very small sample and therefore with lower precision, a new 14C dating is always better than no 14C dating. These conclusions are intuitive for all of us, but this paper has the merit of demonstrating it. That's why I would recommend publication pending some improvements. Indeed, the paper fails to show that results are independent of the software (OxCal) used and of the way to consider the floating chronology. Clear

description of differences between OxCal's models (V-Sequence, T_Sequence ….) and the rationale behind the choice are also missing. This will be useful for all readers who are not familiar with OxCal. A test considering another chronological software (such as Bacon or BCal amongst others) should strengthen the demonstration. It is no clear to me why authors chose to work with constant uncertainties instead of real measurement uncertainties. These comments and others are gathered on the manuscript itself (provided as supplement thereafter).

Please also note the supplement to this comment:
https://www.geochronology-discuss.net/gchron-2019-19/gchron-2019-19-EC1-supplement.pdf

**Supplement:**

[revised manuscript text omitted]

---

## Author Comment (AC1) · 4 Mar 2020

*Response: Thank you for the review and comments on our manuscript. Our response to the comments are interjected in italic font below.*

**SUMMARY The manuscript describes the results of a case study in which radiocarbon ages obtained using gas-source technique are compared with radiocarbon ages of the conventional graphitized samples; both types of samples come from a number of selected depth intervals in a lake sediment core. Because this**

[Figure]

core supposedly has a relatively well resolved varve-based chronology (albeit floating and not shown in this manuscript), the authors integrate varve counts and two types of radiocarbon ages into a simulated 'best age estimate' model. They further demonstrate a series of exercises in generating the synthetic age-depth distributions with a purpose of illustrating the effect of sampling density and sample size (mass carbon) on age model precision. According to the authors, the main idea of the work is an evaluation of how reliable gas source radiocarbon ages on miniature samples are for constructing age models. This is important for those lake records which lack enough datable material for the conventional radiocarbon analyses. The implications highlighted by the authors include (a) how to "improve sampling strategies" (the more age determinations the better, just as one may expect); and (b) what are the "expectations of age uncertainty". Among the benefits of skipping the graphitization step when using gas-source technique the authors cite "reduced cost", but there is no comparison provided for the respective costs for the two types of the techniques used.

NOTES The manuscript leans excessively toward theoretical evaluations of 'how things would be if: : :' and misses a discussion of several key points, which are named but not explored: depositional lags, outlier dates, examples of sample size effect on the radiocarbon date uncertainty as applied to a real core. This happens because the authors chose to (a) treat all their dates as equally good/likely; (b) use the 'best age estimate' for the sequence using everything at once, that is, they combined 3 varve count series + miniature+ regular + graphitized + gas source radiocarbon dates to make a single 'best age estimate'. No wonder there are no outliers if all these things are bundled together.

*The reviewer is correct: our discussion of outlier ages and depositional lags is rather theoretical, but this is simply due to the lack of evidence for outliers in our dataset. From our point of view, it is notable that all these types of age information agree without any outliers that don't fit with the other age information. We find no statistically significant*

*evidence that any single age is an outlier. All ages from within a single level have overlapping 95% confidence intervals. We can envision several ways to test whether the varve count agrees with the $^{14}C$ ages. However because the varve count is floating, all of these methods rely on $^{14}C$ ages in some way, which does restrict our ability to detect outliers. If we tie the varve count to the combined $^{14}C$ ages of the uppermost dated level (732.5 cm), we find that the 95% confidence intervals of all $^{14}C$ ages overlap with the varve count age estimate when we consider the uncertainty of the varve count as well as the uncertainty of the age of the tie point. However, the selection of this tie point is arbitrary. Another approach is to use least squares minimization to minimize the offset between the varve count and all of the $^{14}C$ ages. We did this using the median calibrated age from combined $^{14}C$ ages within each dated level. We find that the 95% confidence intervals of every $^{14}C$ age overlap with the varve count (in this case they overlap even without considering the uncertainty of the varve-count-based age estimate). Please see the attached figure to see how these methods compare. Our OxCal V-sequence 'best-age estimate' yields the same result – all $^{14}C$ ages overlap with the median of this age-model. We will revise the manuscript to emphasize that the floating varve chronology is consistent with all of the radiocarbon ages.*

**As a reader, going from the Introduction to Discussion I expected to see the Figures showing step by step how overlapping varve-based chronologies look like first and how their cumulative error changes with depth, then how a certain number of graphitized regular ages help improving these chronologies and errors, and then how adding gas-source ages on the regular-size samples improves this chronology further, and then how adding gas-source ages on the less reliable miniature samples may or may not improve it even further. Instead, I see a single red line as 'best age estimate' from the very start and then 9 software-generated arbitrary age-depth scenarios. One does not need a sediment core to generate these latter graphs.**

*We thank the reviewer for the suggestion to include a figure that shows the varve count*

*directly compared to the radiocarbon ages; we plan to include the attached figure in the revised manuscript. The suggestion to reorganize the study to start with varve counts, then use graphitized ages to improve the varve count, and then add gas-source ages takes the study in a different direction than we had intended. This would seem to help answer a specific question - how does adding 31 miniature gas-source radiocarbon ages improve an existing chronology based on varve counts and graphitized radiocarbon ages? However, most sediment cores will not be sampled and dated in the same way as our core, and we are using the varve counts to help develop the chronology rather than using the $^{14}C$ ages to check the varve counts. Our goal was to address more widely applicable questions about the tradeoffs between the number of radiocarbon ages and their precision. We do, however, show in Figure 2A how age models constructed using only gas-source ages or only graphitized ages compare to an age model with all radiocarbon ages, which is along the lines of the reviewer's suggestion. Finally, while it is true that one could simulate age-depth scenarios without a sediment core, our simulations are directly informed by empirical data from our core, providing a direct connection to real-world application, which we feel is valuable.*

**Depositional lags for organic fragments are discussed in a purely theoretical way. There appear to be three different varve chronologies, why not show each one of them and see which dates support which one (if any)? A test for potential age outliers would be more robust in this case.**

*The three varve counts are replications by 3 different people which we use to establish the master varve chronology and its uncertainty. The uncertainty is determined based on the agreement of all 3 counts for each individual layer using methods described by Bonk et al. 2015, and Åżarczyński et al. 2018. With the uncertainty range shown, it is redundant to plot or discuss each replicate count individually. Additionally, when viewing a plot of the 3 counts it can be difficult to assess the extent to which the counts agree at the scale of lamina, which is critical to the count uncertainty. It is possible for two counts to include the same number of total varves, but disagree about the location*

*of those varves. This type of disagreement is included in our uncertainty estimate, but is difficult to observe in a plot.*

**Supposedly, as admitted by the authors, the younger the portion of the studied sequence, the more robust is the varve-based chronology. Why not take advantage of this and have a closer look at the potential depositional lags in the most reliable upper portion of the record?**

*We are uncertain of the exact meaning of this comment. The reviewer might be suggesting that we take a closer look at the potential depositional lags in the portion of the core published in Bonk et al. (2015) and Ążarczyński et al. (2018), which we briefly discuss this in lines 304-311. Alternatively, the reviewer might be suggesting that we obtain more ages from this section of the core, which would move the study toward the topic of depositional lags. However, the problem of depositional lags has been considered by previous studies, and is not the main focus of this manuscript, though it is somewhat relevant in that greater dating density (enabled by gas-source) may assist with detecting outliers.*

**What if the varve-only age models were used to compare with gas only and/or graphite only ages? The importance of mass for the reliability of the dates is stressed a number of times by the authors, but their Figures are not informative enough to illustrate this. For example, when discussing age offsets, why not show symbols of different size somehow proportionate to sample mass in Figure 3 and provide respective error bars for each of the dating points?**

*We thank the reviewer for the suggestion about Figure 3. This will be modified to show symbols of different size based on the mass of the sample. Including error bars would make the figure rather cluttered, and essentially equivalent information can be gleaned from Figure 2B.*

**If the sample mass is so important for the age date and bigger is definitely better (as shown in Fig.1), then is it really a good approach to consider all the dates**

**equal in constructing the 'best age estimate'?**

*It is true that bigger sample masses yield more precise dates. However, in our view, that alone is not a reason for removing an age from an age-depth model. We believe it is best to consider all dates as equally valid rather than removing dates without very strong reasons for doing so. Large analytical uncertainty does not indicate a date is invalid or unhelpful. The age modelling routines take into account the differences in precision through the use of probability density functions and thus give more weight to the more precise ages.*

**If the authors found room for nine simulated graphs in the manuscript, I think it would be beneficial to see two-three age-depth graphs using best dates, small-sample dates, and then all dates for comparison.**

*Age-depth models using gas-source ages (miniature samples), graphitized ages, and all ages are already included in Figure 2A.*

**The section 4.4 "Recommendations: : :" is a disappointment as it states a number of trivial basic things about radiocarbon dating, which can be found anywhere and which are not supported by the data the authors present. For example: "we are convinced: : : that miniature samples: : : are better than bulk" – convinced based on what? There is no data presented to support this level of certainty. Indeed, it would have been a really nice test if they were to analyze at least couple bulk samples from the same horizons to see how they compare with those on sieved fragments. "Dating small amounts: : : is preferable to pooling : : :", "a rule of thumb is: : :" – again, there are no data in the paper supporting this conclusion. It appears that these didactic statements are pasted from elsewhere.**

*These are fair points: most of our recommendations are based on existing literature rather than data in this paper. We plan to revise this section to make the reasoning more clear. Even if some of these recommendations come from sources outside the*

Interactive
comment

*data in this paper, they are relevant to the topic of the paper – the usefulness of minia-*
*ture sample masses for radiocarbon dating.*

**"If ages do not agree well : : : youngest ae is most likely to be correct" - what about applying this principle to their own data set and showing how it works out in their studied portion of the lake record? It appears that in the paper the authors cite, Bonk et al. (2015) did just that and identified a number of outliers.**

*The key to the quoted recommendation is "If age results do not agree well...". In our case, the ages agree within the expected uncertainty. We cannot say that the older ages in our dataset are older due to a depositional lag, they are likely older due to the random variation expected with radiocarbon measurements (as defined by the analytical uncertainty). The difference between the Bonk et al 2015 study and ours is that their radiocarbon ages are generally more precise (larger sample mass), and their varve chronology is linked to the surface, which greatly reduces uncertainty and allows for easier detection of outlying $^{14}C$ ages.*

**Finally, the argument of 'cost reduction' for gas-source ages as compared to graphitized samples is used a number of times in the manuscript. Indeed, costs are lab-specific, however, it would be of interest to have at least some estimate in % since the authors repeatedly bring this issue up themselves.**

*We will revise section 4.4 to read as follows: Injecting CO2 into the AMS rather than generating graphite and packing a target substantially reduces the effort to analyze a sample following pre-treatment. Each sample also spends less time on the AMS when introduced as gas rather than graphite. These advantages are partly offset by the additional attention needed during gas source measurements. How these differences translate to per-sample costs depends on the pricing structures implemented in each lab. Cost estimates from two MICADAS labs at the University of Bern and Northern Arizona University range between around 15 and 33% lower costs for gas-source measurements compared to graphitized samples.*

**I suggest substantial revisions, not "major" but at the same time not "minor" or technical either.**

**Fig. 1.** All radiocarbon ages and their 95% calibrated uncertainties plotted versus the varve count. The gray bands show the varve count tied to the combined calibrated age of the uppermost 14C ages (at 732.5

---

## Author Comment (AC2) · 4 Mar 2020

*Response: Thank you for the helpful comments and suggestions to improve manuscript. We have responded to comments in the pdf of the manuscript (see supplement). Here we address the most important comments in more detail.*

**This paper intends to show that a chronology of a sequence is all the better constrained the more dates it contains. It also aims at showing that even on the basis of a very small sample and therefore with lower precision, a new $^{14}$C**

**dating is always better than no $^{14}$C dating. These conclusions are intuitive for all of us, but this paper has the merit of demonstrating it. That's why I would recommend publication pending some improvements. Indeed, the paper fails to show that results are independent of the software (OxCal) used and of the way to consider the floating chronology. Clear description of differences between Ox-Cal's models (V-Sequence, T_Sequence. . .) and the rationale behind the choice are also missing. This will be useful for all readers who are not familiar with Ox-Cal. A test considering another chronological software (such as Bacon or BCal amongst others) should strengthen the demonstration. It is no clear to me why authors chose to work with constant uncertainties instead of real measurement uncertainties. These comments and others are gathered on the manuscript itself (provided as supplement thereafter).**

*We will revise the manuscript to expand on the description of the two OxCal sequences used (P-Sequence and V-Sequence) and the rationale behind those choices. We feel the OxCal V-sequence is the best available tool to integrate varve count data with radiocarbon ages into a single age estimate with uncertainty. We have considered other techniques could be used to assign calendar ages to the floating varve chronology. One could choose a dated level within the core and tie the varve count to the radiocarbon based age at this level. A disadvantage of this technique is the assumption that the tie point age is correct (not subject to contamination or depositional lag). Additionally, when considering the uncertainty of the tie point age and the varve count uncertainty, the varve chronology would have very large errors (Figure ECR1). Another considered method is to use least squares minimization to fit the floating varve count to all of the radiocarbon ages. This technique yields very similar results to the OxCal V-sequence, verifying that the best-age estimate result is not dependent on the choice of statistical routine. Figure ECR1 will be added to the manuscript to more clearly show how the varve counts relate to the $^{14}$C ages and the V-sequence best age-estimate. The OxCal V-sequence is preferable to other techniques because all uncertainties are incorporated into the statistical model, and it allows for the possibility that master varve count*

*may include errors (within the counting uncertainty).*

*Regarding the choice of age-modelling software, we expect that the key conclusions and results of the study are not dependent on the chronological software. Each modelling software may yield slightly different results (or even the same software can easily yield different results if one uses different parameters). However, the general patterns that are essential to the conclusions of the paper (e.g. more ages yield better age models) are expected to hold true for any widely used Bayesian age-depth modelling routine. To demonstrate that our results are not dependent on the use of OxCal, we created age-depth models using Bacon (Blaauw and Christen, 2018) for one iteration of the simulated radiocarbon dating scenarios. We set the acc.rate=8 (average accumulation time through the section) and thickness = 5 cm, all other parameters were the default setting. The results from the Bacon models are highly similar to the models produced using the same ages in OxCal. See Figures below.*

*Finally, the comment about using measurement uncertainties rather than a constant age uncertainty in $^{14}C$ years BP is well taken. We chose to work with a constant uncertainty for simplicity, and because over the period of our studied section (2.1-6.8 ka), the effect of age on uncertainty is relatively small. However, we recognize that the effect of sample age on the uncertainty is important, and should be mentioned in the manuscript. In the revised version of the manuscript we will clarify and emphasize that not only mass affects radiocarbon age uncertainties, but also the age of the sample. We will also include information about our measurement uncertainties in $F^{14}C$. We have modified Figure 1 to include 2 versions of the plot- one with uncertainty in years, and one with uncertainty in $F^{14}C$. Through this figure readers can see expected uncertainty in years (for samples ranging in age from 2000-7000 cal BP), which is more intuitive for readers who are not accustomed to working with $F^{14}C$ values. Additionally, the more widely applicable $F^{14}C$ values are also given for radiocarbon experts or those working on older samples.*

*Figure Captions:*

*Figure ECR1: All radiocarbon ages and their 95% calibrated uncertainties plotted versus the varve count. The gray bands show the varve count tied to the combined calibrated age of the uppermost $^{14}C$ ages (at 732.5 cm) with dark grey indicating the uncertainty calculated from the three replicated varve counts and light gray representing the uncertainty of the tie point. Dashed green is the varve count fit to the $^{14}C$ ages using least squares minimization of the offset between the varve age and the calibrated combined $^{14}C$ ages at each sampled depth.*

*Figure ECR2: Revised Figure 1 from submitted manuscript to demonstrate the relationship between sample mass C and age uncertainty in in terms of years and $F^{14}C$.*

*Figure ECR3: OxCal age-depth models of simulated ages (iteration 2).*

*Figure ECR4: Bacon age-depth models of simulated ages (iteration 2).*

*Figure ECR5: Same as Figure 5 in the submitted manuscript, with overlay of results from Bacon models using the synthetic ages from one single iteration (plotted as red squares). The OxCal results from the same iteration of synthetic ages are plotted as red circles. We propose to include this as a supplemental figure attached to the manuscript to demonstrate that the results are not dependent on the choice of software.*

Depth (cm)

- ◆ Graphitized age
- ◯ Gas-source age
- Master varve count (tied to 14C ages at 732.5)
- Varve count uncertainty
- Error from calibrated 14C ages at tiepoint
- - - Master varve count (fit to all 14C ages using least-squares minimization)
- Median age from OxCal V_seq

Age (cal yr BP)

**Fig. 1.** Figure ECR1

A

350

300

250

Measurement uncertainty 1σ (years)

200

150

100

50

0

40 µg C

○ Graphite  ◆ Gas Source

—— Power Model
(Fit to data)

y = 879.421x$^{-0.502}$
R² = 0.904

Carbon mass (µg)

10          100          1000

B

0.035

0.030

0.025

Measurement uncertainty 1σ (F14C)

0.020

0.015

0.010

0.005

0.000

40 µg C

—— Power Model
(Fit to data)

y = 0.067x$^{-0.496}$
R² = 0.872

Carbon mass (µg)

10          100          1000

**Fig. 2.** Figure ECR2

**5 ages, 35 µg**

ZAB best-age estimate
Simulated P-seq median
Simulated P-seq 95% CI
Simulated 14C age
(median calibrated age
and 95% CI)

**10 ages, 35 µg**

**20 ages, 35 µg** # 2

**5 ages, 90 µg**

**10 ages, 90 µg**

**20 ages, 90 µg**

**5 ages, 500 µg**

**10 ages, 500 µg**

**20 ages, 500 µg**

**Fig. 3.** Figure ECR3

**Fig. 4.** Figure ECR4

[Figure]

A

[Figure]

Fig. 5. Figure ECR5

**Supplement:**

[revised manuscript text omitted]

---

## Author Response (AR2)

**Response to Associate Editor Decision**

Dear authors,

Many thanks for the revised manuscript. You took into consideration both reviewers' comments and this improved the original text. There are still some "technical to minor" changes that need to be addressed before I'll be pleased to accept your paper for publication.

*Response: Thank you for your comments to improve our manuscript, and thank you for accepting the manuscript for publication. Line numbers in the responses below refer to the previously submitted revised manuscript.*

• " For instance, we will mention that expected uncertainty of a 35µg C sample is ± 0.0114 F14C, which translates to 148 14C years if the sample is 4000 years old, but would be smaller for younger samples and larger for older samples." This is a good idea but I'm not able to see it in the revised manuscript. Please consider to add it somewhere (even in Figure 1 caption)

*We actually added the following to the revised manuscript (sorry, forgot to change the wording in the Response to your comments): Line 24: "For samples larger than 40 µg C and younger than 6000 yr BP, the uncalibrated 1σ age uncertainty is consistently less than 150 years (± 0.010 F$^{14}$C)."*

*And Line 475: "Holocene samples containing greater than 40 µg C produce $^{14}$C measurements with analytical uncertainty expected to be less than ± 0.01 F$^{14}$C (150 years for samples than are approximately 4000 years old). Uncertainty increases exponentially as samples get smaller so 10 µg C samples are expected to have uncertainty of ± 0.021 F$^{14}$C (277 years)."*

*We have further clarified the matter by revising line 188 to read "For a sample with 35 µg C, we expect a measurement uncertainty of ± 148 years (or ± 0.0114 F$^{14}$C), which is representative for the average age of all samples in this study (approximately 4000 $^{14}$C yr BP). In reality, older samples would have greater age uncertainty, while younger samples would have less uncertainty. However, the effect of these differences on the performance of simulated age-depth models would be minimal as roughly half the ages would be more precise and half would be less precise." We have also revised Table 2 to include the expected measurement uncertainty in F$^{14}$C for the different sample masses we used in simulations.*

• Figure 1 caption: please complete the caption by adding the range of ages this figure is drawn for

*The caption now includes the statement: "Note that these samples date to approximately 2000-6000 BP; older ages will have greater age uncertainties."*

• I missed the point on "if age results do not agree well, the youngest age is most likely to be correct (assuming no contamination by modern carbon)." raised by reviewer #1. There is no rationale behind that assertion. It is possible to contaminate both young and old samples in lab and young and old samples are both subject to rewording in the core. => please remove or provide me with serious arguments.

*This statement has been removed.*

• Each sample also spends less time on the AMS when introduced as gas rather than graphite. => to be corrected. This is not true in the current state. It is true that we avoid graphitization step that in this way reduce the treatment time. If sample is introduced through EA-GIS interface, it requires a bit more presence of the AMS pilot than if introduced through the solid interface and if it is introduced through cracking GIS, it means much more time for the pilot around the machine as he

has to adjust source tuning for each sample. You toned down a bit your recommendation by adding that point but arguing that the time in the machine is shorter is not at the same level. By the way, it is not clear to me which interface you use (EA or cracking). Furthermore, during the chemical step, working with ultra-small sample is much more time-consuming than with large samples. We have to very careful not to lose any fragment during each rinsing step and be sure it takes time and we definitively prefer to work with large samples. You can say as short as you did, that working with gas source spares time. That's not true. We remove one step but we spend much more time on the remaining steps (in chemistry and on the machine itself). Please adapt.

The message is that now with MICADAS we are able to run small samples and it is better to run a small sample than a big one that would not be well constrained (because it covers more than one varve or because it is bulk) and thanks of that it is now possible to get independent dating for more levels that it previously was. That's sufficient. I'm definitively not sure you need to push argument on time or on cost.

*We will remove the sentence "Each sample also spends less time on the AMS when introduced as gas rather than graphite." from Line 462. We also agree that the chemical cleaning is typically more time-consuming with miniature samples, and have revised Line 468 to say, "
[revised manuscript text omitted]

[Figure]

Figure 2

[Figure]

Figure 3

[Figure]

685

Figure 4

[Figure]

Figure 5

[Figure]

Figure 6

[Figure]

Table 1

<table>
<tr><th>Lab ID</th><th>Core ID</th><th>Top Core Depth (cm)</th><th>Bottom Core Depth (cm)</th><th>Centered Composite Depth (cm)</th><th>Carbon Mass (μg)</th><th>Gas/ Graphite</th><th>$^{14}$C age (BP)</th><th>Calibrated Age (Cal yr BP)[1]</th><th>Modelled Age from OxCal V-sequence (Cal yr BP)[2]</th><th>Material</th></tr>
<tr><td>BE-9791.1.1</td><td>ZAB-12-4-3-2</td><td>75</td><td>77</td><td>732.5</td><td>168</td><td>Gas</td><td>2028 ± 72</td><td>1823-2293</td><td>2106-2218</td><td>*Pinus sylvestris* seed fragments (seed wing, and fragments of seed)</td></tr>
<tr><td>BE-9793.1.1</td><td>ZAB-12-4-3-2</td><td>75</td><td>77</td><td>732.5</td><td>34</td><td>Gas</td><td>2149 ± 112</td><td>1867-2361</td><td>2106-2218</td><td>Terrestrial seed fragment</td></tr>
<tr><td>BE-9792.1.1</td><td>ZAB-12-4-3-2</td><td>75</td><td>77</td><td>732.5</td><td>11</td><td>Gas</td><td>2190 ± 322</td><td>1416-2968</td><td>2106-2218</td><td>Periderm (coniferous)</td></tr>
<tr><td>BE-9794.1.1</td><td>ZAB-12-4-3-2</td><td>75</td><td>77</td><td>732.5</td><td>11</td><td>Gas</td><td>2386 ± 328</td><td>1636-3325</td><td>2106-2218</td><td>Woody scale</td></tr>
<tr><td>BE-9503.1.1</td><td>ZAB-12-3-4-2</td><td>36</td><td>37</td><td>762</td><td>36</td><td>Gas</td><td>2273 ± 117</td><td>1998-2702</td><td>2297-2402</td><td>*Alnus* seed fragments</td></tr>
<tr><td>BE-9502.1.2</td><td>ZAB-12-3-4-2</td><td>85</td><td>86</td><td>811</td><td>87</td><td>Gas</td><td>2358 ± 84</td><td>2159-2715</td><td>2611-2703</td><td>Dicotyledonous leaf fragment[3]</td></tr>
<tr><td>BE-9502.1.1</td><td>ZAB-12-3-4-2</td><td>85</td><td>86</td><td>811</td><td>127</td><td>Gas</td><td>2379 ± 82</td><td>2183-2722</td><td>2611-2703</td><td>Dicotyledonous leaf fragment[3]</td></tr>
<tr><td>BE-9501.1.1</td><td>ZAB-12-3-4-2</td><td>85</td><td>86</td><td>811</td><td>21</td><td>Gas</td><td>2809 ± 201</td><td>2437-3447</td><td>2611-2703</td><td>Deciduous tree/shrub woody scales</td></tr>
<tr><td>BE-9500.1.1</td><td>ZAB-12-3-4-2</td><td>85</td><td>86</td><td>811</td><td>553</td><td>Graphite</td><td>2544 ± 41</td><td>2490-2754</td><td>2611-2703</td><td>Dicotyledonous leaf fragments, woody scales</td></tr>
<tr><td>BE-9497.1.1</td><td>ZAB-12-4-4-2</td><td>20</td><td>21</td><td>861</td><td>131</td><td>Graphite</td><td>2799 ± 67</td><td>2760-3076</td><td>2850-2929</td><td>*Pinus sylvestris* needle</td></tr>
<tr><td>BE-9498.1.1</td><td>ZAB-12-4-4-2</td><td>20</td><td>21</td><td>861</td><td>120</td><td>Graphite</td><td>2820 ± 72</td><td>2774-3143</td><td>2850-2929</td><td>Woody scale</td></tr>
<tr><td>BE-9496.1.1</td><td>ZAB-12-4-4-2</td><td>20</td><td>21</td><td>861</td><td>115</td><td>Graphite</td><td>2857 ± 73</td><td>2790-3174</td><td>2850-2929</td><td>*Pinus sylvestris* needle</td></tr>
<tr><td>BE-9499.1.1</td><td>ZAB-12-4-4-2</td><td>20</td><td>21</td><td>861</td><td>120</td><td>Graphite</td><td>2885 ± 72</td><td>2807-3229</td><td>2850-2929</td><td>Periderm (deciduous)</td></tr>
<tr><td>BE-9495.1.1</td><td>ZAB-12-4-4-2</td><td>61.5</td><td>62.5</td><td>902.5</td><td>21</td><td>Gas</td><td>3158 ± 252</td><td>2764-3984</td><td>3113-3187</td><td>Periderm, Dicotyledonous leaf fragments, woody scales</td></tr>
<tr><td>BE-9494.1.1</td><td>ZAB-12-4-4-2</td><td>61.5</td><td>62.5</td><td>902.5</td><td>54</td><td>Gas</td><td>2845 ± 96</td><td>2761-3215</td><td>3113-3187</td><td>Dicotyledonous leaf fragment</td></tr>
<tr><td>BE-9494.1.2</td><td>ZAB-12-4-4-2</td><td>61.5</td><td>62.5</td><td>902.5</td><td>50</td><td>Gas</td><td>2968 ± 99</td><td>2876-3374</td><td>3113-3187</td><td>Dicotyledonous leaf fragment</td></tr>
<tr><td>BE-9494.1.3</td><td>ZAB-12-4-4-2</td><td>61.5</td><td>62.5</td><td>902.5</td><td>52</td><td>Gas</td><td>2944 ± 97</td><td>2866-3358</td><td>3113-3187</td><td>Dicotyledonous leaf fragment</td></tr>
</table>

| BE-9493.1.1 | ZAB-12-4-4-2 | 61.5 | 62.5 | 902.5 | 230 | Graphite | 2980 ± 56 | 2979-3340 | 3113-3187 | Dicotyledonous leaf fragments, periderm fragments |
|---|---|---|---|---|---|---|---|---|---|---|
| BE-9491.1.1 | ZAB-12-4-4-2 | 100.5 | 101.5 | 941.5 | 37 | Gas | 3197 ± 119 | 3078-3700 | 3391-3462 | Periderm |
| BE-9490.1.2 | ZAB-12-4-4-2 | 100.5 | 101.5 | 941.5 | 123 | Graphite | 3296 ± 74 | 3375-3696 | 3391-3462 | Dicotyledonous leaf fragments |
| BE-9490.1.1 | ZAB-12-4-4-2 | 100.5 | 101.5 | 941.5 | 328 | Graphite | 3145 ± 51 | 3226-3466 | 3391-3462 | Dicotyledonous leaf fragments |
| BE-9489.1.1 | ZAB-12-3-5-2 | 44 | 45 | 1001.4 | 691 | Graphite | 3542 ± 45 | 3697-3965 | 3845-3915 | Dicotyledonous leaf fragments |
| BE-9489.1.2 | ZAB-12-3-5-2 | 44 | 45 | 1001.4 | 179 | Graphite | 3593 ± 62 | 3717-4084 | 3845-3915 | Dicotyledonous leaf fragment[3] |
| BE-9489.1.4 | ZAB-12-3-5-2 | 44 | 45 | 1001.4 | 222 | Graphite | 3603 ± 59 | 3724-4086 | 3845-3915 | Dicotyledonous leaf fragment[3] |
| BE-9489.1.3 | ZAB-12-3-5-2 | 44 | 45 | 1001.4 | 182 | Graphite | 3616 ± 62 | 3725-4141 | 3845-3915 | Dicotyledonous leaf fragment[3] |
| BE-9489.1.5 | ZAB-12-3-5-2 | 44 | 45 | 1001.4 | 124 | Graphite | 3631 ± 75 | 3721-4153 | 3845-3915 | Dicotyledonous leaf fragment[3] |
| BE-9795.1.1 | ZAB-12-4-5-1 | 24 | 26 | 1031.2 | 42 | Gas | 3724 ± 107 | 3829-4417 | 4084-4155 | *Betula* seed fragments, terrestrial woody material, woody scale, periderm fragments |
| BE-9487.1.1 | ZAB-12-4-5-1 | 25 | 26 | 1031.7 | 23 | Gas | 3856 ± 194 | 3731-4832 | 4084-4155 | Leaf fragments |
| BE-9488.1.1 | ZAB-12-4-5-1 | 25 | 26 | 1031.7 | 22 | Gas | 3856 ± 203 | 3725-4836 | 4084-4155 | Wood fragment, Periderm fragments |
| BE-9485.1.1 | ZAB-12-4-5-1 | 75 | 76 | 1081.7 | 60 | Gas | 4062 ± 97 | 4296-4837 | 4540-4616 | Periderm, woody scales |
| BE-9486.1.1 | ZAB-12-4-5-1 | 75 | 76 | 1081.7 | 46 | Gas | 4042 ± 105 | 4249-4832 | 4540-4616 | *Betula alba* seed |
| BE-9484.1.1 | ZAB-12-4-5-1 | 75 | 76 | 1081.7 | 266 | Graphite | 4065 ± 52 | 4421-4813 | 4540-4616 | Periderm fragments |
| BE-9483.1.2 | ZAB-12-4-5-1 | 118.5 | 119.5 | 1125.2 | 49 | Gas | 4387 ± 108 | 4655-5318 | 4960-5042 | Periderm fragments |
| BE-9483.1.1 | ZAB-12-4-5-1 | 118.5 | 119.5 | 1125.2 | 135 | Gas | 4475 ± 90 | 4860-5434 | 4960-5042 | Periderm fragments |
| BE-9481.1.1 | ZAB-12-5-6-1 | 54 | 55.5 | 1176.1 | 95 | Gas | 4850 ± 104 | 5321-5887 | 5500-5591 | Woody seed fragments, leaf fragments, woody scales |
| BE-9482.1.1 | ZAB-12-5-6-1 | 54 | 55.5 | 1176.1 | 22 | Gas | 5246 ± 232 | 5485-6536 | 5500-5591 | Periderm fragments |
| BE-9480.1.1 | ZAB-12-5-6-1 | 79 | 80 | 1200.8 | 35 | Gas | 5081 ± 228 | 5320-6315 | 5745-5832 | Periderm fragments |
| BE-9479.1.1 | ZAB-12-5-6-1 | 79 | 80 | 1200.8 | 42 | Gas | 5063 ± 127 | 5586-6178 | 5745-5832 | Periderm, woody scale |

| BE-9478.1.1 | ZAB-12-5-6-1 | 79 | 80 | 1200.8 | 111 | Graphite | $5197 \pm 86$ | 5745-6190 | 5745-5832 | Periderm fragments and woody scales |
| BE-9476.1.1 | ZAB-12-5-6-2 | 5 | 6 | 1242.5 | 49 | Gas | $5601 \pm 125$ | 6032-6718 | 6175-6267 | Periderm and woody scale |
| BE-9475.1.1 | ZAB-12-5-6-2 | 5 | 6 | 1242.5 | 72 | Gas | $5294 \pm 107$ | 5768-6300 | 6175-6267 | Periderm |
| BE-9477.1.1 | ZAB-12-5-6-2 | 5 | 6 | 1242.5 | 45 | Gas | $5410 \pm 127$ | 5920-6439 | 6175-6267 | Periderm fragments |
| BE-9474.1.1 | ZAB-12-5-6-2 | 5 | 6 | 1242.5 | 504 | Graphite | $5402 \pm 43$ | 6020-6294 | 6175-6267 | *Pinus* periderm fragments |
| BE-9473.1.3 | ZAB-12-5-6-2 | 45.5 | 46.5 | 1283 | 34 | Gas | $5988 \pm 162$ | 6479-7250 | 6531-6643 | Dicotyledonous leaf fragments |
| BE-9473.1.2 | ZAB-12-5-6-2 | 45.5 | 46.5 | 1283 | 55 | Gas | $5787 \pm 119$ | 6317-6880 | 6531-6643 | Dicotyledonous leaf fragments |
| BE-9473.1.1 | ZAB-12-5-6-2 | 45.5 | 46.5 | 1283 | 74 | Gas | $5868 \pm 107$ | 6415-6949 | 6531-6643 | Dicotyledonous leaf fragments |
| BE-9473.1.4 | ZAB-12-5-6-2 | 45.5 | 46.5 | 1283 | 38 | Gas | $5936 \pm 150$ | 6436-7165 | 6531-6643 | Dicotyledonous leaf fragments |
| BE-9472.1.1 | ZAB-12-5-6-2 | 45.5 | 46.5 | 1283 | 143 | Graphite | $5916 \pm 78$ | 6547-6946 | 6531-6643 | Dicotyledonous leaf fragments, periderm fragment |

[1] Ages calibrated using OxCal 4.3 with the IntCal13 calibration curve (Bronk Ramsey, 2009; Reimer et al., 2013). The range reported here represents the 95% confidence interval.

[2] Range represents 95% confidence interval.

[3] These samples were subsampled from a single fragment prior to analysis, thus samples within the same depth with this symbol have the same true age.

Table 2

| Sample Mass (µg) | Expected Uncertainty (yr)[1] | Expected Uncertainty ($F^{14}C$) | Number of ages in model | | |
| --- | --- | --- | --- | --- | --- |
| | | | 5 ages (1.07 per kyr) | 10 ages (2.14 per kyr) | 20 ages (4.27 per kyr) |
| | | | *Mean 95% CI width (yr)* | | |
| 35 | ± 148 | ± 0.011 | 633 | 527 | 433 |
| 90 | ± 92 | ± 0.007 | 577 | 430 | 335 |
| 500 | ± 39 | ± 0.003 | 524 | 325 | 219 |
| | | | *Mean absolute deviation from OxCal V-sequence (yr)* | | |
| 35 | ± 148 | ± 0.011 | 144 | 99 | 78 |
| 90 | ± 92 | ± 0.007 | 98 | 64 | 65 |
| 500 | ± 39 | ± 0.003 | 42 | 40 | 49 |
| | | | *Chron Score* | | |
| 35 | ± 148 | ± 0.011 | 2.46 | 3.14 | 3.48 |
| 90 | ± 92 | ± 0.007 | 2.87 | 3.64 | 4.09 |
| 500 | ± 39 | ± 0.003 | 3.92 | 4.74 | 5.18 |

710 [1] Expected age uncertainty for an approximately 4000-year-old sample used to inform age-depth model simulations